# TOWARDS ROBUST REFERRING EXPRESSION SEGMENTATION FOR COMPLEX REASONING IN THE WILD

## ABSTRACT

Despite the advances in Referring Expression Segmentation (RES) benchmarks, their evaluation protocols remain constrained, primarily focusing on either single targets with short queries (containing minimal attributes) or multiple targets from distinctly different queries on a single domain. This limitation significantly hinders the assessment of more complex reasoning capabilities in RES models. We introduce WildRES, a novel benchmark that incorporates long queries with diverse attributes and non-distinctive queries for multiple targets. This benchmark spans diverse application domains, thus enabling more rigorous evaluation of complex reasoning capabilities in real-world settings. Our analysis reveals that existing RES models demonstrate substantial performance deterioration when evaluated on WildRES. To address this challenge, we introduce SynRES, an automated pipeline generating densely paired compositional synthetic training data through three innovations: (1) a dense caption-driven synthesis for attribute-rich image-mask-expression triplets, (2) reliable semantic alignment mechanisms rectifying caption-pseudo mask inconsistencies via Image-Text Aligned Grouping, and (3) domain-aware augmentations incorporating mosaic composition and superclass replacement to emphasize generalization ability and distinguishing attributes over object categories. Experimental results demonstrate that models trained with SynRES achieve consistent improvements on not only our complex WildRES benchmark but also classic RES benchmarks (e.g. RefCOCO/+/g). Code is available at this link. Dataset will be available upon acceptance.

## 1 INTRODUCTION

Recent advancements in Large Multimodal Models (LMMs) (Liu et al., 2024b;a) and foundational segmentation models (Kirillov et al., 2023; Ravi et al., 2025) have significantly enhanced language-based image segmentation by improving both open vocabulary capability and sentence understanding for Referring Expression Segmentation (RES) (Lai et al., 2024; Xia et al., 2024; Ren et al., 2024; Rasheed et al., 2024; Zhang et al., 2024b; Chen et al., 2024; Zhang et al., 2024a). These models surpass the constraints of traditional RES approaches (Wang et al., 2022; Liu et al., 2023a; Yang et al., 2022b; Zhao et al., 2023b; Xu et al., 2023; Zou et al., 2023), which rely on completed image-mask-expression triplet datasets (Yu et al., 2016; Mao et al., 2016; Liu et al., 2023a), by leveraging diverse data sources such as semantic segmentation (Caesar et al., 2018; Zhou et al., 2017; Neuhold et al., 2017; Chen et al., 2014; He et al., 2022), referring image segmentation (Yu et al., 2016; Mao et al., 2016; Liu et al., 2023a), and visual question answering (VQA) (Liu et al., 2024b; Antol et al., 2015; Liu et al., 2024a). This broader data integration allows these models to effectively interpret diverse expressions and operate robustly in real-world scenarios.

Furthermore, numerous RES benchmarks (Rohrbach et al., 2016; Yu et al., 2016; Mao et al., 2016; Liu et al., 2023a) have emerged to evaluate models' compositional reasoning and generalization capabilities. Notably, gRefCOCO (Liu et al., 2023a) introduces multi-target or no-target expressions to further assess models' advanced reasoning abilities. Nevertheless, existing benchmarks predominantly concentrate on expressions with minimal attributes (Yu et al., 2016; Rohrbach et al., 2016) or distinctive (countable) expressions (Liu et al., 2023a) within the homogeneous domain of COCO (Kazemzadeh et al., 2014), as illustrated in Fig. 1-(a. 1, 5). This narrow focus substantially constrains the assessment of complex reasoning capabilities required in real-world scenarios. In this paper, we introduce **WildRES**, a new test benchmark designed to reflect advanced reasoning and real-world complexity. WildRES incorporates three new elements as shown in Fig. 1-(a. 2-4, 6-8): (1)

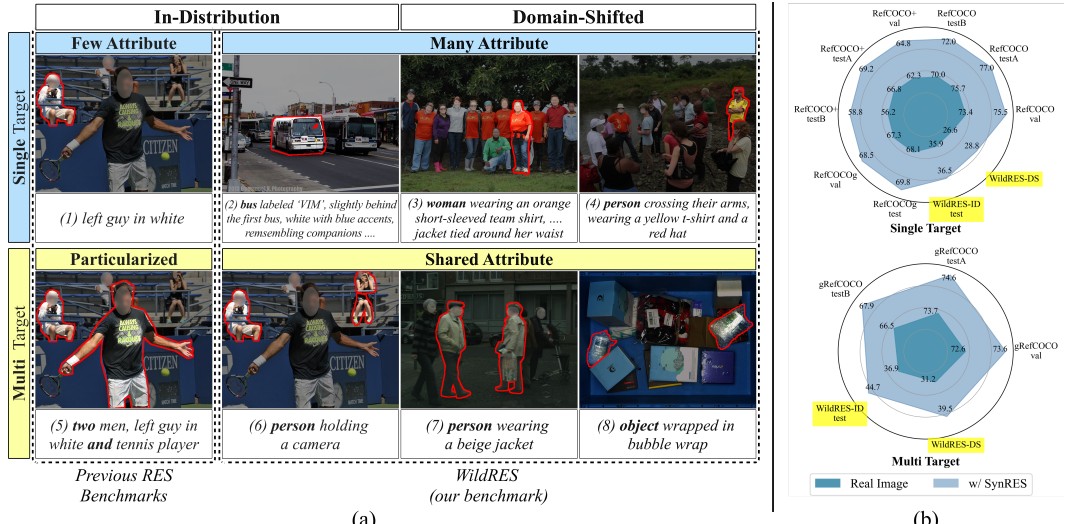

(a)                                                      (b)

Figure 1: (a) Comparison of existing referring expression segmentation (RES) benchmarks with our proposed benchmark, WildRES, which demands advanced complex reasoning and scene understanding. Previous RES datasets (Yu et al., 2016; Liu et al., 2023a) primarily emphasize single-target focus with short queries (1) or distinctive multi-target queries (5) within a similar domain (*e.g*, images from COCO (Caesar et al., 2018)). These constraints limit the evaluation of complex queries and generalization ability of RES models. We propose a new benchmark WildRES, which improves single-target expressions with diverse attributes (2)-(4) and refines multi-target ones with shared traits (non-distinctive) and concise phrasing (6)-(8). In addition, WildRES includes in-distribution (col. 2) and domain-shifted subsets (cols. 3, 4) to evaluate generalizability. (b) The state-of-the-art RES method (LISA (Lai et al., 2024)) experiences notable performance degradation on our benchmark (highlighted in yellow), which requires advanced reasoning and generalization capability. Our novel synthetic data generation, SynRES, enhances the model's compositional reasoning ability.

explicit longer queries containing many attributes, (2) shared (non-distinctive) attribute expressions for multiple targets, and (3) domain-shifted images from challenging environments including densely populated scenes (Shao et al., 2018), autonomous driving contexts (Cordts et al., 2016), and robotic applications (Mitash et al., 2023). Fig. 1-(b) shows the existing RES model (*e.g*, Lai et al. (2024)) exhibits substantial performance deterioration when evaluated on WildRES.

A natural approach to improving real-world RES is to acquire large-scale training data featuring comparable complex expressions and domain-shifted imagery as seen in WildRES. However, manual annotation at this scale is prohibitively costly. For example, crowdsourced datasets like RefCOCO tend to contain minimal expressions sufficient only for target identification (Fig 1 (a-1)). To address this limitation, we adopt a transfer learning strategy leveraging automatically generated synthetic data with compositional queries. Prior works (Wu et al., 2023b; Nguyen et al., 2023; Yang et al., 2023; Ye et al., 2024) generate synthetic segmentation data conditioned on target masks but lack referring expressions required for RES. Pseudo-RIS (Yu et al., 2024) produces synthetic captions and pseudo-masks from SAM (Ravi et al., 2025) trained on COCO (Kazemzadeh et al., 2014), yet does not cater specifically to multi-target or ambiguous queries, resulting in insufficient dense image-expression pairs. Thus, fundamental questions remain regarding how to effectively generate and optimally utilize synthetic data for complex, attribute-rich queries across diverse domains.

We propose **SynRES**, which automatically generates densely paired synthetic referring expressions with many attributes and corresponding images and masks triplets via generative models. As illustrated in Fig. 3, SynRES produces images containing identical objects with heterogeneous attributes, which will subsequently be utilized to train models capable of discriminating between various attribute combinations. SynRES consists of three key steps: 1) generating distinctive synthetic expressions with different attributes for the same object and their corresponding images, 2) grouping semantically similar expressions and aligning them with corresponding synthetic masks, and 3) domain-aware image augmentation and semantic text augmentation to reduce domain gaps in each modality. In step 1, we utilize real images and masks along with image captioning models to create

distinctive captions (Yu et al., 2024). Unlike minimal human expressions these synthetic expressions incorporate rich attributes to improve performance under challenging scenarios (see Fig. 5). These concatenated expressions are then used as input for Text-to-Image (T2I) models to generate synthetic images encompassing all expression features. Step 2 addresses potential inaccuracies in pseudo-masks of the generated images. To resolve such issues, we group related expressions and create segmentation masks aligned with these groups. Finally, in step 3, we enhance the synthetic image-expression-mask triplets through data augmentation techniques tailored for RES in the wild. These include mosaic augmentation for multi-target scenarios and superclass replacement text semantic augmentation to focus on detailed target descriptions rather than single-word references.

Integrating our proposed SynRES yields substantial performance gains for referring expression segmentation (RES) across diverse model architectures and benchmarks. For instance, on the in-domain WildRES-ID benchmark, SynRES improves LISA by +2.0 to +2.8 gIoU, with similar gains observed for GSVA and GLaMM. Furthermore, the method boosts cross-domain generalization, improving performance on the domain-shifted WildRES-DS benchmark by 3.8–6.2 gIoU and consistently outperforming existing baselines. Finally, our comprehensive ablation studies demonstrate that both the textual and visual augmentation components of SynRES are critical to its success.

Our contributions are summarized as follows:

1. We propose WildRES, a novel benchmark for Referring Expression Segmentation in real-world scenarios, covering single-target many-attribute cases and multi-target shared-attribute. WildRES includes diverse domain.
2. We introduce SynRES, an automated pipeline for generating densely paired synthetic datasets with diverse attributes and precise pseudo-masks, enabling effective data augmentation without manual annotation.
3. Experiments demonstrate that SynRES is model-agnostic and enhances existing RES models, showing improvements in wild scenarios and classic benchmarks.

## 2 RELATED WORK

**Referring Expression Segmentation with Large Multimodal Models.** The introduction of LISA (Lai et al., 2024), which leveraged the visual-language capabilities of Large Multimodal Models (LMMs) for the Referring Expression Segmentation (RES) task using <SEG> tokens, has significantly influenced the development of advanced LMM-based RES models (Xia et al., 2024; Ren et al., 2024; Rasheed et al., 2024; Zhang et al., 2024b; Chen et al., 2024; Zhang et al., 2024a; Wu et al., 2024; 2025). Notably, GSVA (Xia et al., 2024) and PixelLM (Ren et al., 2024) have improved multi-target segmentation accuracy by employing strategies such as multiple <SEG> tokens or introducing target refinement loss. To train these models effectively, large-scale datasets from diverse domains are utilized. Examples include semantic segmentation datasets (Caesar et al., 2018; Zhou et al., 2017; Neuhold et al., 2017; Chen et al., 2014; He et al., 2022), classic RES datasets (Yu et al., 2016; Mao et al., 2016), and Visual Question Answering datasets (Liu et al., 2024b; Antol et al., 2015; Liu et al., 2024a). These datasets are adapted for RES tasks using prompt-based transformations or integrated into training pipelines. Such adaptations enhance vocabulary comprehension and reasoning capabilities. Some models employ specialized datasets tailored for generalized RES tasks. For instance, GSVA (Xia et al., 2024) and SAM4LMM (Chen et al., 2024) incorporate gRefCOCO (Liu et al., 2023a) into their training data. Other notable datasets include ReasonSeg (Lai et al., 2024) and GranD (Rasheed et al., 2024), which are derived from automatically annotated SA-1B (Kirillov et al., 2023) or GranD-f (based on Flickr-30K (Young et al., 2014), RefCOCOg (Mao et al., 2016), and PSG (Yang et al., 2022a)). Existing models exhibit performance degradation when evaluated on our more challenging benchmark, WildRES. Our model-agnostic approach, SynRES, significantly enhances the complex reasoning capabilities of these models.

**Synthetic Data and Augmentation for Referring Expression Segmentation.** Early efforts on synthetic datasets for segmentation leveraged GANs (Zhang et al., 2021; Li et al., 2022). With the advent of diffusion models (Ho et al., 2020; Song et al., 2021), recent methods generate class-targeted images paired with pseudo-segmentation masks (Nguyen et al., 2023; Yang et al., 2023), and also synthesize masks with corresponding high-fidelity images (Ye et al., 2024), extending to instance and panoptic settings (Zhao et al., 2023a; Fan et al., 2024; Xie et al., 2024b; Tu et al., 2025). However, these approaches predominantly yield image–mask pairs and do not explicitly align

textual descriptions with mask semantics. In parallel, multimodal data augmentation has advanced joint image–text learning (Hao et al., 2023; Liu et al., 2023b; Jin et al., 2024; Wu et al., 2023a), while RES-specific augmentation remains comparatively underexplored. NeMo (Ha et al., 2024) forms mosaics from semantically related but non-redundant images, and Pseudo-RIS (Yu et al., 2024) leverages pseudo masks and diverse captions to enable weakly supervised training. Yet these techniques only partially capture the interplay among image content, referring expressions, and segmentation masks, which is critical for RES. We address this gap by generating densely aligned image–mask–text triplets—covering the same objects with diverse attributes and reliable pseudo-masks—and by introducing augmentation strategies that jointly operate on all three modalities to reduce domain discrepancies, enforce semantic consistency across transformations, and improve generalization in RES.

## 3 Referring Expression Segmentation in the Wild

### 3.1 Problem Definition

In RES, given an input image $\mathbf{x}_{img}$ and an input language query $\mathbf{x}_{txt}$, the goal is to produce the corresponding binary segmentation mask $\mathbf{M}$. In addition to this traditional RES, we aim to address the three key aspects: i) using referring expressions with diverse attributes to pinpoint a single precise target in Fig. 1-(a, 2-4), ii) identifying specific multiple targets with shared attributes in Fig. 1-(a, 6-8), and iii) extending these two aspects to other domains, as demonstrated in Fig. 1-(a, 3, 4, 7, 8). For multi-target scenarios, we select images containing at least three objects and design the referring expressions so that only a subset of these objects satisfies the specified attributes. Detailed descriptions emphasize unique characteristics of individual objects, compelling segmentation models to discern both shared and distinct features.

### 3.2 WildRES: A New RES Benchmark for Advanced Reasoning and Generalization

We introduce WildRES, an evaluation benchmark designed to evaluate segmentation models in complex, real-world contexts with 724 images and 941 expressions. WildRES does not contain the training set, only the validation and test set. WildRES includes both in-distribution images (WildRES-ID) sourced from MSCOCO (Caesar et al., 2018) and domain-shifted images (WildRES-DS) representing unseen scenes from datasets such as CrowdHuman (Shao et al., 2018), Cityscapes (Cordts et al., 2016), and ARMBench (Mitash et al., 2023), all aimed at assessing model robustness in challenging environments.

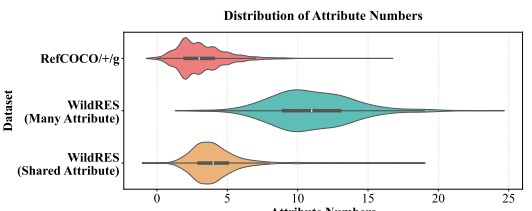

Figure 2: Number of attributions in classic RES datasets vs. WildRES. Using GPT-4o-mini (Achiam et al., 2023), we counted the number of attributes shown in Table C. Single-target expressions in WildRES often exceed 11 attributes to specify a target, while multi-target expressions have fewer attributes, similar to classic RES datasets.

WildRES is divided into two subsets: Many Attribute (MA) and Shared Attribute (SA). MA subsets comprise images that contain at least 5 objects, with each caption manually reviewed to ensure precise alignment with its corresponding segmentation mask. SA subsets focus on images featuring multiple, referable objects from the same class that exhibit distinct attributes, thereby evaluating fine-grained segmentation capabilities. In WildRES-ID, both subsets are present; in contrast, within WildRES-DS, datasets are categorized according to their dominant features—CrowdHuman, which contains numerous individuals, corresponds to the Many Attribute subset, while Cityscapes and ARMBench, featuring multiple objects with shared attributes, align with the Shared Attribute subset. The exact quantity for each image and expression constituting WildRES is described in detail in Appendix A.

Existing RES datasets often lack diversity in their attributes. To systematically analyze this limitation, we categorized referring expressions into 8 distinct attributions (more details of each attributions are in Appendix B). Utilizing GPT-4o-mini (Achiam et al., 2023), we classified referring expressions from these datasets into the predefined categories. Fig. 2 illustrates the results of this analysis. We observed that expressions from traditional RES datasets are composed of simple and easily targetable

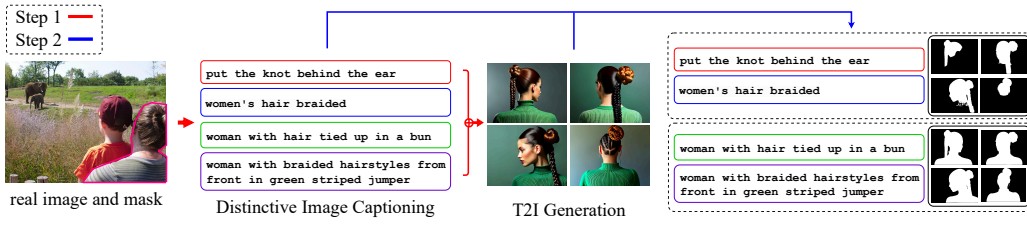

Figure 3: Overview of the step 1 and 2 in SynRES. The process begins by creating distinctive $n$ synthetic expressions for target objects from real images and masks. These expressions are then concatenated and input into a pre-trained text-to-image generative model to produce $m$ synthetic images. Finally, a grouping step is performed to generate reliable synthetic masks by associating appropriate $l$ segmentation masks with their corresponding expressions, yielding densely paired image-mask-expression triplets for objects with diverse attributes, thereby facilitating the learning of distinctive attribute combinations across objects.

samples, with a mean attribute count below 4. In contrast, WildRES's Many Attribute expressions exceed an average of 11 attributes, while maintaining a consistent number of attributes in expressions when referring to multiple targets sharing attributes. Detailed differences from previous datasets (Liu et al., 2023a; You et al., 2025) are described in Appendix C.

## 4 OUR METHOD: SYNRES

We propose SynRES, a fully automated synthetic dataset generation and augmentation pipeline for challenging RES, as shown in Fig. 3. As existing data generation approaches (Yang et al., 2023; Yu et al., 2024) do not produce densely paired image-mask-expression triplets, they could be suboptimal for developing models capable of discriminating between objects based on diverse attribute queries extracted from synthetic data. SynRES generates densely paired distinctive synthetic expressions and their corresponding images in Step 1. We obtain reliable pseudo semantic segmentation masks with image-text aligned grouping in Step 2. Our synthetic dataset is subsequently enhanced with data augmentation techniques to train models with improved generalization capabilities in Step 3.

### 4.1 STEP 1: SYNTHETIC DISTINCTIVE REFERRING EXPRESSIONS AND IMAGE GENERATION

We first aim to generate distinctive synthetic expressions with diverse attributes. Motivated by (Yu et al., 2024), SynRES employs image captioning models to generate distinctive expressions $\{x_{\text{txt},j}^{\text{syn}}\}_{j=1}^{n}$ for individual referring objects from real image $x_{\text{img}}^{\text{real}}$ and it's corresponding mask $M$ as in Fig. 3.

For image synthesis, we construct composite prompts by aggregating the generated expressions to ensure comprehensive feature representation. For example, by combining the expressions "*put the knot behind the ear*" and "*women's hair braided*", the aggregated description "*put the knot behind the ear, women's hair braided*" yields structured inputs with structured template selection per generation instance (see prompt templates in Appendix D). We leverage SANA (Xie et al., 2024a), a text-to-image model that efficiently generates high-fidelity synthetic images $\{x_{\text{img},i}^{\text{syn}}\}_{i=1}^{m}$ with different $m$ seeds while maintaining visual-semantic alignment.

### 4.2 STEP 2: RELIABLE SYNTHETIC MASK GENERATION WITH IMAGE-TEXT ALIGNED GROUPING

T2I models are inherently stochastic, generating $m$ images from identical prompts with different seeds to produce visual variations where not all capture each expression feature. We leverage multi-view consensus by computing mIoU globally across images to aggregate independent observations and filter outliers. Expression pairs exceeding threshold $\tau$ form consensus groups whose masks are refined through averaging, yielding robust pseudo-masks.

After the step 1, distinctive synthetic expression may or may not align with identical synthetic masks. This process establishes aligned groupings between synthetic images $\{x_{\text{img},i}^{\text{syn}}\}_{i=1}^{m}$ and textual

expressions $\{\mathbf{x}^{\text{syn}}_{\text{txt},j}\}^n_{j=1}$ through semantic-aware mask consensus. This validation process generates refined segmentation masks $\mathbf{M}^{\text{syn}}_{(i,j)}$ for each semantically aligned group as shown in Fig. 4.

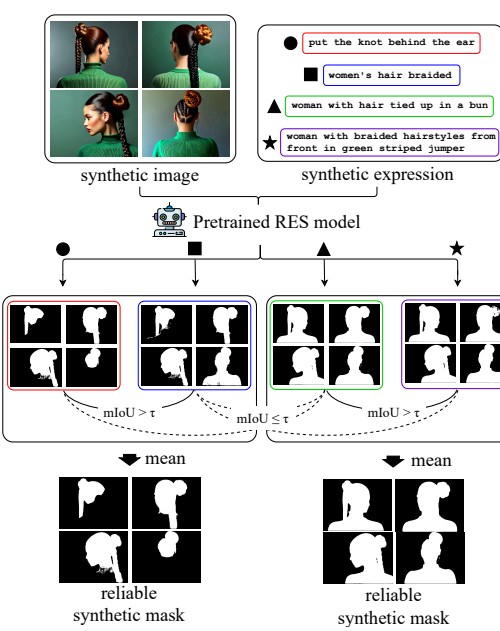

Figure 4: Synthetic images and texts are aligned via pseudo mask generation, binary conversion, and pairwise IoU-based clustering into consensus groups (same-colored rectangles represent text-inferred pseudo masks from synthetic images). Final refined masks are computed by per-group averaging and thresholding, ensuring high-quality alignment.

First, we generate $n$ pseudo segmentation masks $\hat{\mathbf{M}}^{\text{psd}}_{(i,j)}$ for each synthetic image $\mathbf{x}^{\text{syn}}_{\text{img},i}$ using a pretrained RES model (Lai et al., 2024), producing $m \times n$ candidate masks. An identification function $\mathbb{I}$ converts continuous mask values into binary masks through confidence thresholding:

$$\mathbf{M}^{\text{psd}}_{(i,j)} = \mathbb{I}\left(\hat{\mathbf{M}}^{\text{psd}}_{(i,j)}\right) = \begin{cases} 1 & \text{if } \hat{\mathbf{M}}^{\text{psd}}_{(i,j)} \geq 0.5 \\ 0 & \text{otherwise} \end{cases}.$$

This per-image mask generation ensures high-quality pseudo segmentation masks construction as each synthetic image mostly contains only one primary object (Ha et al., 2024) and at the same time, pretrained RES models are good at generate the masks of a single target.

Next, we compute pairwise Intersection-over-Union (IoU) scores between all binary mask pairs $\mathbf{M}^{\text{psd}}_{(i,j_1)}$ and $\mathbf{M}^{\text{psd}}_{(i,j_2)}$ derived from different textual expressions for the same image. The mean IoU (mIoU) across all images for each expression pair is:

$$\text{mIoU}(j_1, j_2) = \frac{1}{m} \sum^m_{i=1} \text{IoU}\left(\mathbf{M}^{\text{psd}}_{(i,j_1)}, \mathbf{M}^{\text{psd}}_{(i,j_2)}\right).$$

Expression pairs achieving mIoU scores exceeding threshold $\tau$ are clustered into consensus groups $G_k$, while singleton expressions below this threshold are discarded. We then compute refined masks through per-group averaging:

$$\hat{\mathbf{M}}^{\text{syn}}_{(i,j)} = \frac{1}{|G_k|} \sum_{j \in G_k} \hat{\mathbf{M}}^{\text{psd}}_{(i,j)},$$

where $G_k$ denotes the consensus group containing expression $\mathbf{x}^{\text{syn}}_{\text{txt}}$.

Finally, we apply the identification function to these refined masks:

$$\mathbf{M}^{\text{syn}}_{(i,j)} = \mathbb{I}\left(\hat{\mathbf{M}}^{\text{syn}}_{(i,j)}\right),$$

yielding binary masks where pixels with averaged values $> 0.5$ are assigned 1, and 0 otherwise. For ease of notation, $\mathbf{M}^{\text{syn}}_{(i,j)}$ is denoted $\mathbf{M}^{\text{syn}}_k$ where k ranges from 1 to $l$.

By proceeding these things, we generated synthetic triplets consisting of images, text expressions, and segmentation masks: $\{\mathbf{x}^{\text{syn}}_{\text{img},i}\}^m_{i=1}$, $\{\mathbf{x}^{\text{syn}}_{\text{txt},j}\}^n_{j=1}$, and $\{\mathbf{M}^{\text{syn}}_k\}^l_{k=1}$, respectively. These were derived from a single real image and its corresponding mask, denoted as $\mathbf{x}^{\text{real}}_{\text{img}}$ and $\mathbf{M}^{\text{real}}$ (see Appendix H.1 for before and after image-text aligned grouping images).

### 4.3 STEP 3: DOMAIN-AWARE AUGMENTATION FOR IMAGES-MASKS AND TEXT

Step 3 addresses synthetic data limitations: isolated single-object images lack multi-target complexity, and models overfit to specific categories. Mosaic augmentation creates multi-target scenes while reducing synthetic-real domain gaps. Superclass replacement (e.g., woman→person, probability $p$) shifts focus from categories to attributes, enabling robust vocabulary generalization.

Figure 5: Augmented examples in the step 3. Mosaic augmentation is applied using synthetic images and masks containing one original real image (blue border) and masks. Specific words (*e.g*, woman) are replaced with their superclass (*e.g*, person), a broader concept, with a probability of $p$, which could effectively mitigate model bias toward specific terminology and facilitate the learning of broader vocabulary associations. This substitution process may potentially produce false negative masks (magenta arrows) when other objects belonging to the same superclass exist within the image. Although this frequently exists in human expressions (*e.g*, RefCOCO+), we manually mitigate this challenge by generating isolated single objects within our synthetic data.

**Multi-Target Aimed Image-Mask Augmentation.** In RES in the wild, it is important for a model to distinguish the same objects with different attributes. We apply mosaic augmentation (Xie et al., 2024b; Ha et al., 2024) by constructing composite arrangements with synthetic images and masks to facilitate multi-object/target segmentation as shown in Fig. 5. In this process, each mosaic randomly adopts either a $2 \times 2$ grid (1 real image $\mathbf{x}_{\text{img}}^{\text{real}}$ + 3 synthetic $\mathbf{x}_{\text{img,i}}^{\text{syn}}$) or $3 \times 3$ grid (1 real + 8 synthetic), with corresponding masks $\mathbf{M}^{\text{real}}$ and $\mathbf{M}_i^{\text{syn}}$. This approach facilitates the accommodation of multiple target objects within a single augmented scene. Simultaneously, integrating real and synthetic images creates composite visuals that effectively mitigate the domain gap between real-world and synthetically generated images.

**Debiased Text Augmentation.** We observe that training data commonly demonstrates a significant bias toward specific object class terminology. This systematic bias results in segmentation failures when processing input queries containing alternative synonymous expressions. We apply the text augmentation that probabilistically replaces head and sub-nouns with their corresponding superclasses. For instance, the phrase "*woman with hair tied up in a bun*" is transformed into "*person with hair tied up in a bun*" with probability $p$, thereby shifting the emphasis toward the action and mitigating gender bias (see Appendix E for superclass and original words). This approach is especially useful for our synthetic data, as implementing such augmentation with conventional RES datasets (e.g., RefCOCO) would present significant challenges: these collections typically contain multiple objects and provide only minimal discriminative cues necessary for precise target identification. Consequently, modifying expressions with superclass terminology could inadvertently introduce non-distinctive queries, potentially resulting in imprecise segmentation masks through the inclusion of non-target objects (False negative masks in Fig. 5). In contrast, our method circumvents this issue because our synthetic expressions are richly descriptive (averaging 11.4 words). They contain redundant attributes that preserve sufficient discriminative cues even after superclass replacement.

## 5 EXPERIMENTS

### 5.1 EXPERIMENTAL SETUPS

**Network Architectures.** To demonstrate the effectiveness of our approach, we applied our method to the popular lightweight RES model ReLA Liu et al. (2023a) as well as large multimodal models such as LISA-7B/13B Lai et al. (2024), GSVA-7B Xia et al. (2024), and GLaMM-8B Rasheed et al. (2024). These implementations were selected because they provide fully open-sourced training code, datasets, and pre-trained weights. For other architectural components requiring hyperparameters (*e.g*, projection layer dimensions), we adhered to the configurations specified in the original papers.

**Implementation Details.** We utilized 4 NVIDIA A6000 48G GPUs for training. Other settings, including the optimizer and learning rate scheduler, were consistent with those of the network architecture. Specifically, we generated $m = 6$ synthetic images per referring target and up to $n = 5$ synthetic expressions based on RefCOCO+. This resulted in a total of 129,552 synthetic images,

| Model | Training data | | val | | test | | | | | |
|---|---|---|---|---|---|---|---|---|---|---|
| | | | overall | | many attribute | | shared attribute | | overall | |
| | real | synthetic | gIoU | cIoU | gIoU | cIoU | gIoU | cIoU | gIoU | cIoU |
| *RES Specialized Model* | | | | | | | | | | |
| ReLA | ✓ | ✗ | 30.0 | 33.6 | 15.6 | 15.6 | 36.9 | 37.9 | 25.9 | 28.0 |
| | ✓ | ✓ SynRES (Ours) | **33.9** | **38.1** | **18.2** | **19.9** | **38.0** | **39.8** | **27.8** | **31.7** |
| *LMM based RES Models* | | | | | | | | | | |
| LISA-7B | ✓ | ✗ | 37.1 | 40.3 | **30.1** | 25.0 | 39.3 | 36.0 | 34.5 | 32.5 |
| | ✓ | ✓ FreeMask Yang et al. (2023) | 38.8 | 42.0 | 25.8 | 25.3 | 37.1 | 36.4 | 31.6 | 33.3 |
| | ✓ | ✓ SynRES (Ours) | **41.3** | **46.1** | 29.4 | **26.3** | **43.6** | **43.3** | **36.5** | **38.1** |
| LISA-13B | ✓ | ✗ | 44.0 | 45.1 | 35.9 | 31.6 | 36.9 | 37.8 | 37.7 | 36.0 |
| | ✓ | ✓ SynRES (Ours) | **47.7** | **51.8** | **36.5** | **33.4** | **44.7** | **43.9** | **40.5** | **40.8** |
| GSVA-7B | ✓ | ✗ | 38.8 | 42.8 | **32.6** | 26.1 | 37.7 | 34.8 | 34.8 | 32.0 |
| | ✓ | ✓ SynRES (Ours) | **41.3** | **46.5** | 32.4 | **26.3** | **43.6** | **41.5** | **38.0** | **36.9** |
| GLaMM-8B | ✓ | ✗ | 38.6 | 41.4 | 34.8 | 29.7 | 38.3 | 36.8 | 36.0 | 34.4 |
| | ✓ | ✓ SynRES (Ours) | **42.0** | **44.6** | **38.0** | **31.1** | **38.9** | **38.9** | **39.6** | **37.5** |

Table 1: WildRES-ID segmentation results among SynRES (ours) and baselines.

| Model | Training data | | Domain source | | | | | | average | |
|---|---|---|---|---|---|---|---|---|---|---|
| | | | CrowdHuman | | Cityscapes | | ARMBench | | | |
| | real | synthetic | gIoU | cIoU | gIoU | cIoU | gIoU | cIoU | gIoU | cIoU |
| *RES Specialized Model* | | | | | | | | | | |
| ReLA | ✓ | ✗ | 14.4 | 13.8 | 23.5 | 40.3 | 26.9 | 24.4 | 21.6 | 26.2 |
| | ✓ | ✓ SynRES (Ours) | **21.4** | **20.7** | **25.9** | **40.5** | **28.4** | **27.0** | **25.2** | **29.4** |
| *LMM based RES Models* | | | | | | | | | | |
| LISA-7B | ✓ | ✗ | **25.6** | 26.3 | 37.1 | 39.4 | 26.9 | 26.0 | 29.9 | 30.6 |
| | ✓ | ✓ FreeMask Yang et al. (2023) | 23.5 | 26.0 | 34.4 | 36.9 | 24.5 | 26.1 | 27.5 | 29.7 |
| | ✓ | ✓ SynRES (Ours) | 24.8 | **27.3** | **40.5** | **43.2** | **35.8** | **33.3** | **33.7** | **34.6** |
| LISA-13B | ✓ | ✗ | 26.6 | 29.4 | 34.1 | 34.8 | 28.3 | 26.9 | 29.7 | 30.4 |
| | ✓ | ✓ SynRES (Ours) | **28.8** | **30.0** | **39.9** | **43.6** | **39.1** | **37.7** | **35.9** | **37.1** |
| GSVA-7B | ✓ | ✗ | 19.8 | 24.6 | 35.9 | 36.5 | 25.1 | 23.6 | 26.9 | 28.2 |
| | ✓ | ✓ SynRES (Ours) | **21.4** | **25.5** | **41.2** | **44.7** | **33.6** | **32.6** | **32.1** | **34.2** |
| GLaMM-8B | ✓ | ✗ | 35.0 | 32.2 | 20.0 | 24.4 | 32.4 | 29.3 | 29.1 | 28.6 |
| | ✓ | ✓ SynRES (Ours) | **35.5** | **34.3** | **28.7** | **30.5** | **36.4** | **33.8** | **33.5** | **32.9** |

Table 2: WildRES-DS segmentation results among SynRES (ours) and baselines.

78,263 expressions, and 151,116 masks. The default mIoU threshold $\tau$ in SynRES step 2 was set to 0.65, and the replacement probability $p$ for expression data augmentation was set to 0.7.

**Evaluation Metrics.** We employed two evaluation metrics: gIoU and cIoU. The gIoU metric represents the average IoU values calculated per image, whereas cIoU is computed as the cumulative intersection over the cumulative union. Given cIoU's high sensitivity to large-area objects and higher-resolution images, gIoU was primarily used for evaluation Lai et al. (2024).

5.2 WILDRES: RES IN THE WILD

**Experimental Settings.** Regarding ReLA, since its standard protocol involves training on individual datasets rather than joint training across multiple sources, we focus on gRefCOCO, which features longer expressions and multiple targets compared to RefCOCO(+/g). We compare ReLA trained solely on gRefCOCO against a version trained on both gRefCOCO and SynRES, employing an 8:2 data ratio.

For LISA, GSVA, and GLaMM, we fine-tune the officially released pretrained weights augmented with SynRES. LISA and GSVA use datasets originally designed for semantic segmentation (Zhou et al., 2017; Caesar et al., 2018; Ramanathan et al., 2023; Chen et al., 2014; He et al., 2022), RefCOCO(+/g) (Yu et al., 2016; Rohrbach et al., 2016; Mao et al., 2016), VQA (Liu et al., 2024b;a), and ReasonSeg (for LISA). For GSVA, we additionally incorporate gRefCOCO (Liu et al., 2023a) during fine-tuning. We set a 9:3:3:1:4 data ratio across semantic segmentation, classic RES, VQA, ReasonSeg, and SynRES. For GLaMM, following the original setting, we fine-tune on RefCOCO(+/g) and SynRES with a 3:4 ratio.

All experiments use official pretrained weights for each RES model, training for 5,000 steps with WildRES-ID validation every 100 steps regardless of synthetic data inclusion. Notably, the WildRES-DS remains *unused* during validation.

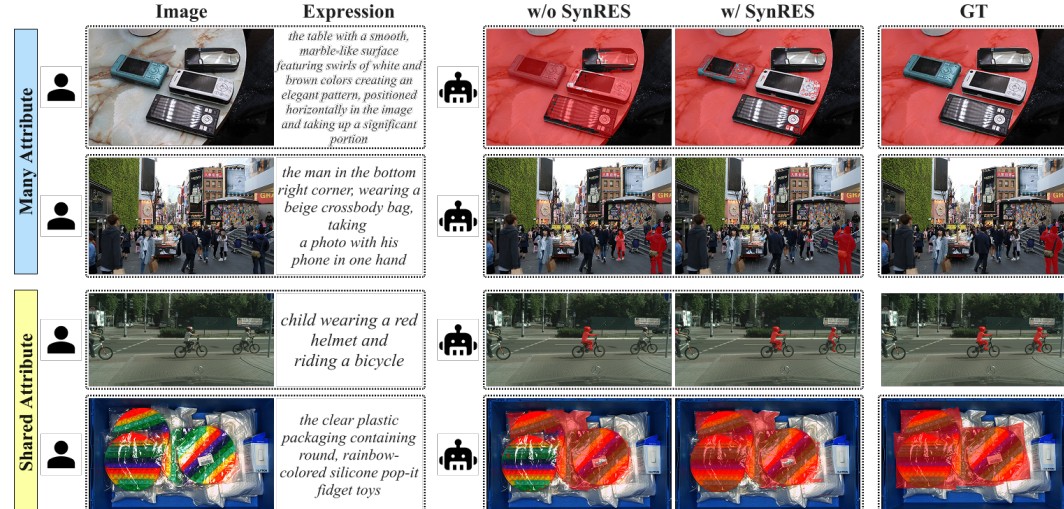

Figure 6: Qualitative results on WildRES, comparing LISA trained w/o and w/ SynRES.

**Results.** Tab. 1 shows that real data with SynRES yields consistent, architecture-agnostic gains on the WildRES-ID validation split, mirrored on the held-out test set (including multi-target and attribute-sharing regimes). With validation used for selection and the test set reserved for final reporting, the improvements reflect genuine generalization rather than split-specific tuning.

Tab. 2 evaluates cross-domain robustness on WildRES-DS with CrowdHuman, Cityscapes (autonomous driving), and ARMBench (robotics), where SynRES-added models outperform real-only baselines across all architectures. Gains are especially pronounced on ARMBench and raise the overall averages, indicating resilience to distribution shifts without reliance on a single source domain.

Results show that simply adding more synthetic data does not solve RES. FreeMask, though beneficial for semantic segmentation with real data, lowered performance across domains on WildRES-DS. In contrast, SynRES consistently improved results, showing that RES requires careful handling of synthetic images, text, and masks—an approach realized by SynRES.

We compared the output masks of the LISA fine-tuned with SynRES and the LISA without SynRES on the WildRES dataset, as shown in Fig. 6. The visualization shows that fine-tuning the model with SynRES reduces false negatives by capturing ground truth objects in multi-target scenarios involving shared attributes. Additionally, it prevents false positives by accurately understanding the meaning of each words in cases with many attribute expressions.

## 5.3 ADDITIONAL ANALYSIS

### 5.3.1 WILDRES VS. EXISTING RES BENCHMARKS

Under identical protocols, WildRES presents substantially greater challenges than canonical RES benchmarks. Models achieve only 20–50 gIoU on WildRES, a stark contrast to the 60–80 gIoU on single-target RefCOCO/+/g and 70–80 gIoU on multi-target gRefCOCO and RefZOM. This performance gap persists even when applying SynRES, which indicates that WildRES demands more advanced capabilities—specifically, fine-grained disambiguation in multi-target scenes and robust attribute grounding—that are not fully captured by prior benchmarks. For detailed results across different architectures and datasets, see Appendix F.

### 5.3.2 ABLATION STUDY

Tab. 3 presents an ablation study evaluating each component in SynRES. The results show a gradual decline in performance, with the gIoU decreasing from 41.3 to 40.8 and 35.5 when text augmentation and mosaic augmentation were excluded, respectively. Notably, when both vision and text augmentations were omitted, the performance fell below the baseline value of 37.1, which was obtained without training on the synthetic dataset. This performance degradation can be attributed to two primary factors: first, the absence of vision augmentation restricts the model to the Wild RES task, which lacks multiple targets, leading to poor outcomes in the Shared Attribute portion; second, a domain gap between synthetic and real images further degrades performance.

Moreover, the performance improvement with text augmentation confirms superclass replacement's effectiveness for contextual expression understanding in complex scenes by focusing on the description of the target. These findings underscore the importance of both text and vision augmentation in enhancing the effectiveness of the SynRES pipeline for complex visual grounding tasks. Additionally, adopting only domain-aware augmentation without mask refinement in SynRES step 2 shows limited performance, proving the necessity of refining raw pseudo masks.

| Model w/ LISA-7B | Modifications | | | WildRES-ID val | |
|---|---|---|---|---|---|
| | Step 2 | Mosaic Aug. | Text Aug. | gIoU | cIoU |
| SynRES | ✓ | ✓ | ✓ | 41.3 | 46.1 |
| | ✓ | ✓ | ✗ | 40.1 | 44.0 |
| | ✗ | ✓ | ✓ | 39.1 | 41.9 |
| | ✓ | ✗ | ✗ | 35.5 | 38.3 |
| Only Real | ✗ | ✗ | ✗ | 37.1 | 40.3 |

Table 3: Ablation study on the core designs of SynRES. ✓ means the employment of the component while ✗ means not.

### 5.3.3 GENDER BIAS MITIGATION

To evaluate bias mitigation, we conduct counterfactual consistency analysis on WildRES using the LISA-7B model. We select samples containing gender-specific terms (*man*, *woman*, *boy*, *girl*) and replace them with gender-neutral alternatives (*person*, *child*). As shown in Table 4, our SynRES-trained model exhibits only 0.4-point variation (33.1 → 32.7), while the baseline shows 1.1-point fluctuation (31.4 → 30.3). This 3× smaller gap demonstrates that our approach successfully reduces dependency on gender cues, relying more on visual attributes.

| SynRES | gender-specific | neutral | Gap (↓) |
|---|---|---|---|
| ✓ | 33.1 | 32.7 | **0.4** |
| ✗ | 31.4 | 30.3 | 1.1 |

Table 4: Gender bias evaluation via counterfactual consistency on WildRES with gender-neutral word substitution.

## 6 CONCLUSION

In this study, we introduce a new RES task aimed at addressing more complex reasoning in the real-world, dynamic scenarios, WildRES. Specifically, we focus on cases where expressions refer to multiple targets with shared attributes settings or require an increased number of attributes to identify a single target within such environments. To benchmark this task, we not only developed WildRES-ID within the same domain as existing RES datasets but also extended its applicability to autonomous driving and robotics domains through the WildRES-DS. Additionally, we propose a method to enhance the performance of existing models in WildRES by leveraging densely paired synthetic image-expression-mask triplet generation and augmentation, SynRES. We hope that our work broadens the scope of language-based segmentation.

## REPRODUCIBILITY STATEMENT

We are committed to ensuring the reproducibility of our work by providing comprehensive documentation, source code, and data. Our full source code is available via the anonymous link provided in the abstract.

**Methodology and Experiments.** Key details of our methodology and experimental setup are located in the main paper. The proposed SynRES pipeline is described in Section 2, while our complete experimental configuration—including model architectures, data ratios, and primary hyperparameters—is outlined in Section 5.1.

**Implementation Details.** Further implementation details are provided in the appendix. This includes statistics for our new WildRES benchmark (Appendix A), specific prompts for the T2I model (Step 1; Appendix D), terms for superclass replacement (Step 3; Appendix E), and a sensitivity analysis of key hyperparameters such as the mask threshold $\tau$ and replacement probability $p$ (Appendix G.1).

**Code and Data Availability.** The currently released code implements our domain-aware augmentation (Step 3) as configured for the LISA. We plan to update the repository with implementations for GSVA and GLaMM in the future. Due to file size constraints, the supplementary material includes the WildRES-ID. The WildRES-DS will be released at a later date.

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

APPENDIX

## A   DETAILS OF WILDRES

Tab. A provides a detailed number of the WildRES dataset, which consists of the Wildseg-ID and Wildseg-DS subsets. The dataset is categorized based on domain, data split (validation/test), and attribute type (Many Attribute or Shared Attribute). In total, WildRES comprises 724 images and 974 expressions across different domains and attribute types.

## B   COUNT REFERRING EXPRESSION ATTRIBUTES

Based on (Yu & Li, 2024), we refine eight expression attributes identified in existing datasets: *head noun*, *sub noun*, *color*, *size*, *absolute location relation*, *relative location relation*, *action*, and *generic attribute*. A detailed example is shown in Tab. C. Using these attributes, we conduct quantitative analyses comparing the attribute distribution patterns between classic datasets and WildRES. This counting process is automated using ChatGPT-4o-mini (Achiam et al., 2023). Tab.  B shows the prompt template used for the counting process.

| Type | Domain | Split | Attribute | Image | Expression |
|------|--------|-------|-----------|-------|------------|
| Wildseg-ID | MSCOCO | Validation | Many Attribute | 100 | 138 |
| | | | Shared Attribute | 104 | 127 |
| | | | Total | 196 | 265 |
| | | Test | Many Attribute | 108 | 133 |
| | | | Shared Attribute | 115 | 124 |
| | | | Total | 215 | 257 |
| Wildseg-DS | CrowdHuman | Test | Many Attribute | 101 | 212 |
| | Cityscapes | | Shared Attribute | 105 | 120 |
| | Armbench | | Shared Attribute | 107 | 120 |
| Total | | | | 724 | 974 |

Table A: Detailed Numbers of WildRES

---

Here's an example analysis:
Example sentence: "Standing on the right side of the image, this individual wears a vibrant red jacket that contrasts sharply against the snowy backdrop. Their black pants provide a stark visual contrast. The person's black beanie covers their head, and they appear to be in a relaxed stance. Their body language suggests a sense of confidence and readiness for outdoor activities."
Example classification:
    "A1 (Head Noun)": ["individual", "person"],
    "A2 (Sub Noun)": ["jacket", "pants", "beanie", "head", "body language", "stance", "backdrop"],
    "A3 (Color)": ["red", "black", "snowy"],
    "A4 (Size)": [],
    "A5 (Absolute Location Relation)": ["right side of the image"],
    "A6 (Relative Location Relation)": ["on"],
    "A7 (Action)": ["standing", "wears", "covers", "appear", "suggests"],
    "A8 (Generic Attribute)": ["vibrant", "relaxed", "confident", "ready"],
    "Total Attributes Count": 7

Now, analyze this new sentence and classify its words into the same attributes: Sentence: "{sentence}"
Please classify into these categories:
1. Head noun: The main noun being described
2. Sub noun: Supporting nouns
3. Color: Color words
4. Size: Size-related words
5. Absolute location: Fixed position words
6. Relative location: Relative position words
7. Action: Action words/verbs
8. Generic attribute: Other descriptive words
Return only a JSON object in this exact format:
{{
    "A1 (Head Noun)": [],
    "A2 (Sub Noun)": [],
    "A3 (Color)": [],
    "A4 (Size)": [],
    "A5 (Absolute Location Relation)": [],
    "A6 (Relative Location Relation)": [],
    "A7 (Action)": [],
    "A8 (Generic Attribute)": [],
    "Total Attributes Count": 0
}}

Table B: Prompt for counting attributes

| ID | Attribute | Word |
|----|-----------|------|
| A1 | head noun | *cat* |
| A2 | sub noun | *bench, boat* |
| A3 | color | *green* |
| A4 | size | *big* |
| A5 | absolute location relation | *the center* |
| A6 | relative location relation | *on, next to, in* |
| A7 | action | *sitting* |
| A8 | generic attribute | *wooden* |

Table C: Types of attributes and their corresponding words in the referring expression, "*the cat sitting on the bench next to big green wooden boat in the center of the image*"

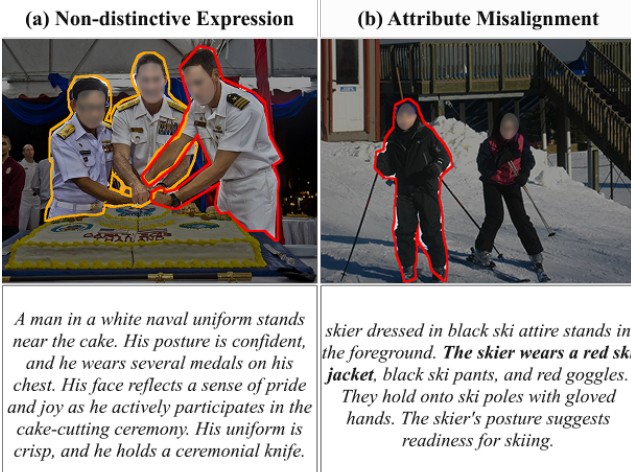

Figure A: Pix2Cap misaligned examples

## C   DIFFERENCES WITH WILDRES AND PREVIOUS DATASETS

### C.1   DIFFERENCES FROM PREVIOUS MULTI-TARGET RES DATASETS

Existing multi-target RES datasets like gRefCOCO (Liu et al., 2023a) and Ref-ZOM (Hu et al., 2023) predominantly rely on enumeration, conjunctions, and plural head nouns. These datasets frequently employ numerical references ("Three persons playing baseball") that assume known referent quantities—an assumption often disconnected from real-world scenarios where shared attributes drive identification.

While gRefCOCO incorporates attribute-based expressions ("A except B" or "A that has B"), these constructions appear infrequently—only 39 and 78 instances in 259,859 expressions—confirming enumeration dominates. Similarly, Ref-ZOM employs template-based generation that combines expressions from one-to-one datasets or embeds category information into predefined structures, sacrificing natural language flexibility for consistency.

WildRES adopts a fundamentally different approach by eliminating explicit numerical references and plural head nouns. Our dataset relies exclusively on shared attributes with singular head nouns to indicate multiple objects, creating more context-driven references. We focus on identifying subsets within categories based on distinctive attributes rather than comprehensive enumeration. For example, "person holding a camera" may coexist with "person not holding a camera" instances, enabling precise differentiation without explicitly counting targets. This design better reflects natural language patterns where speakers rarely enumerate objects explicitly. Unlike gRefCOCO, WildRES excludes no-target expressions entirely.

## C.2 Difference from Pix2Cap

Pix2Cap (You et al., 2025) provides longer and more detailed captions from GPT-4V (Hurst et al., 2024) compared to traditional datasets. However, many captions fail to correspond to their respective objects accurately.

As illustrated in Fig. A, misalignment issues in the Pix2Cap dataset can be categorized into non-distinctive expressions and attribute mismatches. Non-distinctive expressions due to multiple objects (Fig. A (a)) occur when a referring expression corresponds to multiple objects, making the expression non-distinctive. The red-edged mask represents the Pix2Cap annotation, but additional objects matching the same description (highlighted in orange) result in incomplete coverage. Attribute misalignment (Fig. A (b)) arises when certain attributes within a referring expression do not accurately match the corresponding mask. The red mask represents the Pix2Cap annotation for the given expression, but the skier is not wearing a red ski jacket, leading to attribute misalignment.

To address these issues, we refined Pix2Cap-generated expressions or created new ones for images containing three or more objects, constructing an improved dataset with enhanced caption-to-object alignment while preserving Pix2Cap's rich linguistic diversity.

## D  Structured Prompts for T2I Model

We randomly select one of two prompt structures as input for the T2I model in Sec. 4.1:

- *photo of [aggregated description], hyper-realistic, 4k, realism, highly detailed, natural realistic background*
- *cinematic scene [aggregated description], hyper-realistic, 4k, realism, highly detailed, natural realistic background*

These prompts bridge the synthetic-real domain gap while avoiding stylistic elements like cartoon or pixel art

## E  Superclass Replace Words

Tab. D lists superclass replacements for text augmentation. Based on MSCOCO (Caesar et al., 2018) classes and our superclass taxonomy, we select frequently occurring terms from SynRES's synthetic expressions. To maintain expression consistency, gender-specific replacements trigger corresponding pronoun updates. For example, replacing "*boy*" with "*child*" in "*The boy holding his bag*" simultaneously changes "*his*" to "*their*", resulting in "*The child holding their bag*".

## F  Additional Experiments

### F.1  Classic RES

**Experimental Settings.** We leveraged official LISA weights that incorporated the ReasonSeg validation set during training. Performance was evaluated on the ReasonSeg test set for epoch-wise model selection. For GSVA, where official weights exclude the ReasonSeg validation set, we perform model selection using ReasonSeg validation set. Other configurations remain consistent with the previous section.

Since the gRefCOCO was not included during LISA training, there was significant performance degradation on no-target expressions, which are part of gRefCOCO, when evaluated on this dataset. As our goal also included testing multi-target performance on gRefCOCO, no-target expressions were omitted during gRefCOCO evaluation with LISA. In contrast, GSVA utilized the entire gRefCOCO training set, ensuring that all data were equally considered during evaluation.

**Results.** In Tab. E, we observed that using SynRES mostly achieved the best performance across the not only single-target RefCOCO series but also multi-target and no-target included gRefCOCO. Also, incorporating FreeMask (Yang et al., 2023) led to a decline in performance. This outcome highlights

| Idx | Superclass Word | Original Word |
|---|---|---|
| 1 | child | boy, girl, son, daughter |
| 2 | kid | boy, girl, son, daughter |
| 3 | adult | woman, women, man, men, female, male |
| 4 | person | woman, women, man, men, female, male, boy, girl, guy |
| 5 | their | his, her |
| 6 | vehicle | car, bus, plane, train, airplane, truck, boat, motorcycle |
| 7 | animal | bird, cow, bull, rabbit, bunny, dog, puppy, cat, zebra, elephant, horse, giraffe |
| 8 | fruit | apple, banana |
| 9 | vegetable | broccoli, carrot, cabbage, radish |
| 10 | food | sandwich, hot dog, pizza, donut, doughnut, cake, hamburger |
| 11 | electronic | tv, television, laptop, computer, keyboard, cell phone, smartphone |
| 12 | furniture | chair, couch, sofa, bed, desk |

Table D: **Superclass and original words for text augmentation**

| Model | Training data | | RefCOCO | | | RefCOCO+ | | | RefCOCOg | | gRefCOCO | | |
|---|---|---|---|---|---|---|---|---|---|---|---|---|---|
| | real | synthetic | val | testA | testB | val | testA | testB | val(U) | test(U) | val | testA | testB |
| *RES Specialized Model* | | | | | | | | | | | | | |
| CRIS | ✓ | ✗ | 70.5 | 73.2 | 66.1 | 62.3 | 68.1 | 53.7 | 59.9 | 60.4 | 56.3 | 63.4 | 51.8 |
| LAVT | ✓ | ✗ | 72.7 | 75.8 | 68.8 | 62.1 | 68.4 | 55.1 | 61.2 | 62.1 | 58.4 | 65.9 | 55.8 |
| ReLA | ✓ | ✗ | 73.8 | 76.5 | 70.2 | 66.0 | 71.0 | 57.7 | 65.0 | 66.0 | 63.6 | 70.0 | 61.0 |
| X-Decoder | ✓ | ✗ | - | - | - | - | - | - | 64.6 | - | - | - | - |
| SEEM | ✓ | ✗ | - | - | - | - | - | - | 65.7 | - | - | - | - |
| *LMM based RES Models* | | | | | | | | | | | | | |
| F-LMM | ✓ | ✗ | 75.2 | 79.1 | 71.9 | 63.7 | 71.8 | 54.7 | 67.1 | 68.1 | - | - | - |
| VisionLLM v2 | ✓ | ✗ | 76.6 | 79.3 | 74.3 | 64.5 | 69.8 | 61.5 | 70.7 | 71.2 | - | - | - |
| LISA-7B | ✓ | ✗ | 73.4 | 75.7 | 70.0 | 62.3 | 66.8 | 56.2 | 67.3 | 68.1 | - | - | - |
| | ✓ | ✓ FreeMask | 72.4 | 74.8 | 68.3 | 59.7 | 65.3 | 52.9 | 65.4 | 66.0 | - | - | - |
| | ✓ | ✓ SynRES (Ours) | **75.5** | **77.0** | **72.0** | **64.8** | **69.2** | **58.8** | **68.5** | **69.8** | - | - | - |
| GSVA-7B | ✓ | ✗ | **76.0** | **77.8** | **72.7** | 63.6 | 68.2 | 58.3 | 69.9 | 70.7 | 72.6 | 73.7 | 66.5 |
| | ✓ | ✓ SynRES (Ours) | 76.0 | 77.5 | 72.6 | **64.2** | **68.2** | **59.0** | **70.1** | **70.9** | **73.6** | **74.6** | **67.9** |
| GLaMM-7B | ✓ | ✗ | 80.3 | **83.3** | 77.9 | 73.9 | 79.0 | **68.0** | 75.3 | 75.2 | - | - | - |
| | ✓ | ✓ SynRES (Ours) | **80.6** | 83.0 | **78.0** | **74.1** | **79.5** | 67.9 | **75.5** | **75.9** | - | - | - |

Table E: Classic referring segmentation results (gIoU) compared w/ and w/o SynRES on existing RES baselines. All numbers from LISA and GSVA are reproduced using their official repository and weights.

that simply adding synthetic datasets does not guarantee improved performance in RES; instead, it underscores the importance of effectively handling such datasets to achieve positive results.

**Performance Reproduction in LISA.** We reproduced LISA using the official codebase and released weights. Nevertheless, our results were consistently lower than those reported in the paper (e.g., RefCOCO val 74.1), a discrepancy that has been repeatedly noted by other researchers. For example, the GSVA paper reports a RefCOCO val score of 71.7 for LISA-13B—lower than LISA-7B—with similar trends on RefCOCO+ and RefCOCOg. Likewise, Issue #162 on the official LISA GitHub reports RefCOCO val at 72.2 and notes even lower performance for the 7B model. To account for this variability, we conducted three independent runs for LISA and report the median in our tables. In contrast, the results for the remaining models closely matched the numbers reported in their original papers.

| Ref-ZOM test | | | Model | LISA-7B | | LISA-13B | | GSVA-7B | |
|---|---|---|---|---|---|---|---|---|---|
| | gIoU | cIoU | | base | +SynRES | base | +SynRES | base | +SynRES |
| GSVA-7B | 72.58 | 64.40 | val | – | – | – | – | 45.13 | **45.78** |
| GSVA-7B+SynRES | **73.55** | **64.65** | test | **44.66** | 44.13 | 47.62 | **48.02** | 40.19 | **40.72** |

Table F: Results on Ref-ZOM test set (left) and ReasonSeg (right).

## F.2 ADDITIONAL BENCHMARKS

We further conducted experiments on Ref-ZOM (Hu et al., 2023) (multi-target datasets, Tab. F left) and ReasonSeg (Lai et al., 2024) (long and implicit expressions, Tab. F right). Results demonstrate that SynRES maintains or improves baseline performance across diverse scenarios. On Ref-ZOM, GSVA-7B+SynRES preserves strong performance with slight gains of +0.97 gIoU (73.55 vs 72.58). For ReasonSeg, the approach shows consistent stability across model scales: LISA-13B achieves +0.40 gIoU improvement (48.02 vs 47.62), while GSVA-7B maintains competitive performance with +0.65 gIoU on validation and +0.53 gIoU on test data.

## G DISCUSSIONS

### G.1 HYPERPARAMETER CHOICES

| $p$ | gIoU | $\tau$ | gIoU |
|---|---|---|---|
| 0.5 | 40.8 | 0.55 | 40.5 |
| 0.6 | 40.3 | 0.6 | 39.6 |
| 0.7 | **41.3** | 0.65 | **41.3** |
| 0.8 | 41.0 | 0.7 | 40.2 |
| 0.9 | 40.4 | 0.75 | 40.8 |

Table G: Hyperparameter ablation study results with replace probability $p$ and mIoU Threshold $\tau$.

Tab. G presents the results of experiments conducted by varying the replacement probability $p$, a hyperparameter of SynRES, from 0.4 to 0.9. Results demonstrate that the validation set of WildRES-ID consistently achieves gIoU scores above 40 across all configurations, indicating minimal performance sensitivity to $p$ variations. Similarly, when testing the mIoU threshold $\tau$ within 0.55-0.75, the minimum observed gIoU remains above 39.6, showing comparable robustness to $\tau$ selection.

### G.2 PERFORMANCE RELATIVE TO SYNTHETIC DATA QUANTITY

We evaluate LISA's performance using progressively increasing quantities of synthetic training images (25%, 50%, 75%, and 100% of full dataset size). As shown in Tab. H, the gIoU metric demonstrates remarkable stability across data scales, with only 0.6 separating the 25% and 100% conditions. Notably, the 25% configuration achieves 98.5% of the maximum observed performance, suggesting efficient data utilization.

### G.3 COMPUTATIONAL COST

The process of generating the synthetic dataset does not incur additional training costs, as it leverages off-the-shelf models. However, the generation process does require time: generating synthetic expressions took approximately 20 hours, creating synthetic images took around 23 hours, and performing step 2 grouping required about 30 hours, all using four A6000 GPUs. Once the synthetic dataset is generated, effective results can be achieved with only 5,000 steps of additional fine-tuning—just $1/10$ of the training from scratch.

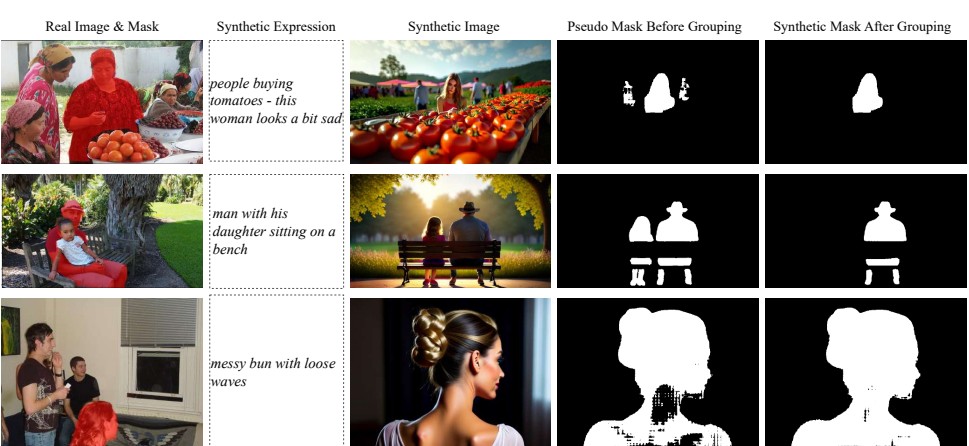

Figure B: Examples of before and after of SynRES step 2.

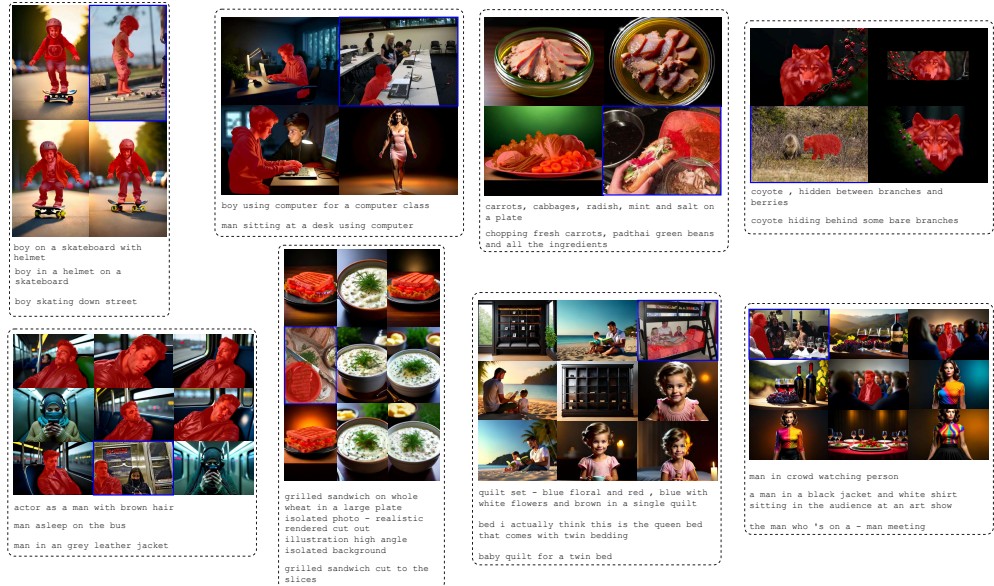

Figure C: Examples of SynRES generated training data. Blue border is real image and mask. Zoom-in for closer look.

| Syn. Images (%) | gIoU |
|---|---|
| 25% | 40.7 |
| 50% | 40.8 |
| 75% | 40.9 |
| 100% | **41.3** |

Table H: Performance variations of LISA with SynRES on WildRES-ID validation set across different quantities of synthetic images.

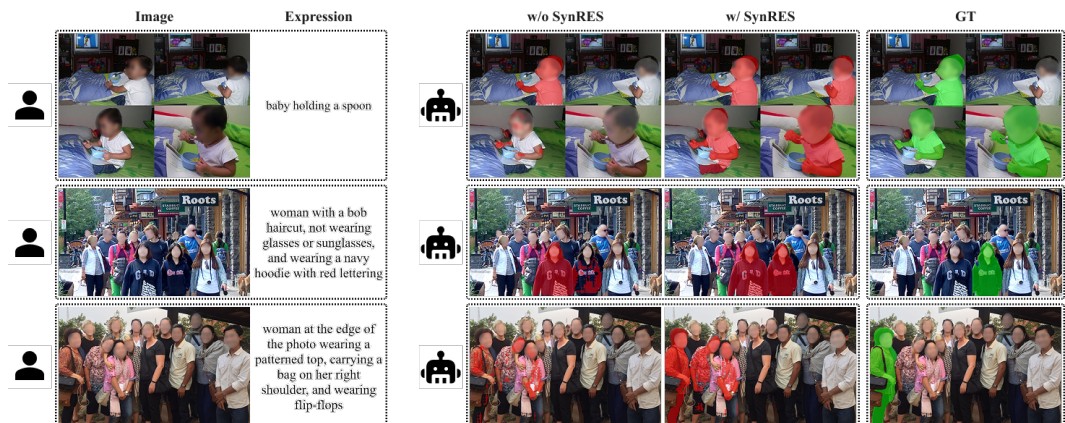

Figure D: Failure Cases

### G.4 LIMITATIONS AND FUTURE WORK

Fig. D demonstrates characteristic failure cases involving over-segmentation beyond ground truth boundaries. These errors typically occur when visually similar objects appear immediately adjacent to targets. Our current mosaic augmentation struggles with such proximate object arrangements and limits composite images to 9 components, restricting training exposure to higher object densities. Implementing targeted copy-and-paste augmentation (Fan et al., 2024) that preserves original attributes could mitigate these limitations in future work.

## H MORE VISUALIZATION

### H.1 REFINED SYNTHETIC MASK EXAMPLE BEFORE AND AFTER SYNRES STEP 2

Fig. B show mask refinement results, comparing synthetic segmentation outputs before and after SynRES in step 2.

### H.2 ADDITIONAL SYNRES EXAMPLES

Fig. C presents SynRES examples demonstrating synthetic dataset generation. Superclass replacement augmentation applied to these generated expressions following Tab. E.

### H.3 EXTENDED QUALITATIVE RESULTS

We present additional visualization results in Fig. E to further demonstrate the effectiveness of the WildRES task with SynRES.

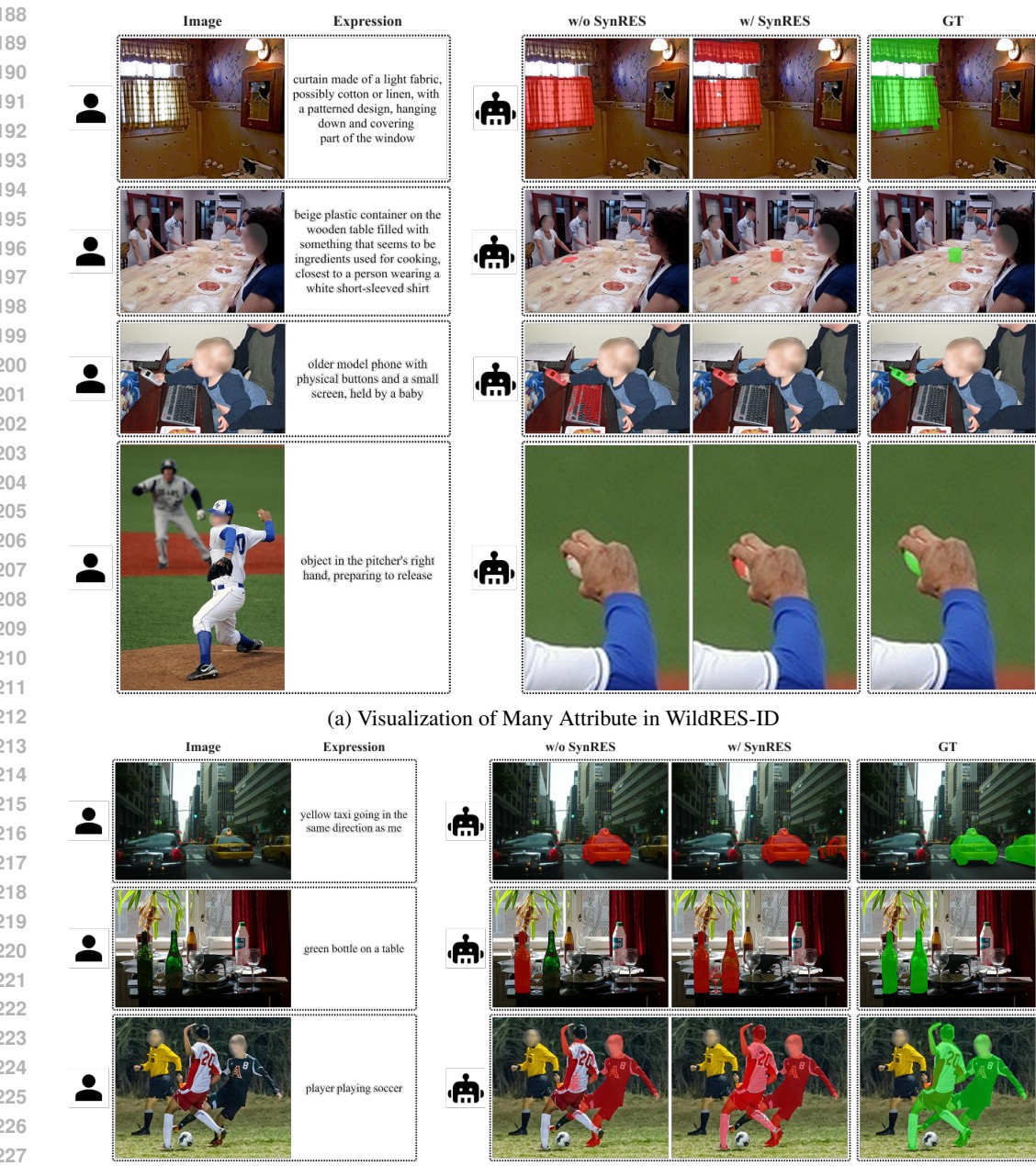

(a) Visualization of Many Attribute in WildRES-ID

(b) Visualization of Shared Attribute in WildRES-ID

# I    LLMs USAGE CLARIFICATION

We used LLMs exclusively for writing assistance and grammar polishing to improve the clarity and presentation of our methods and content. We did not use LLMs for other purposes, such as identifying related work or research ideation.

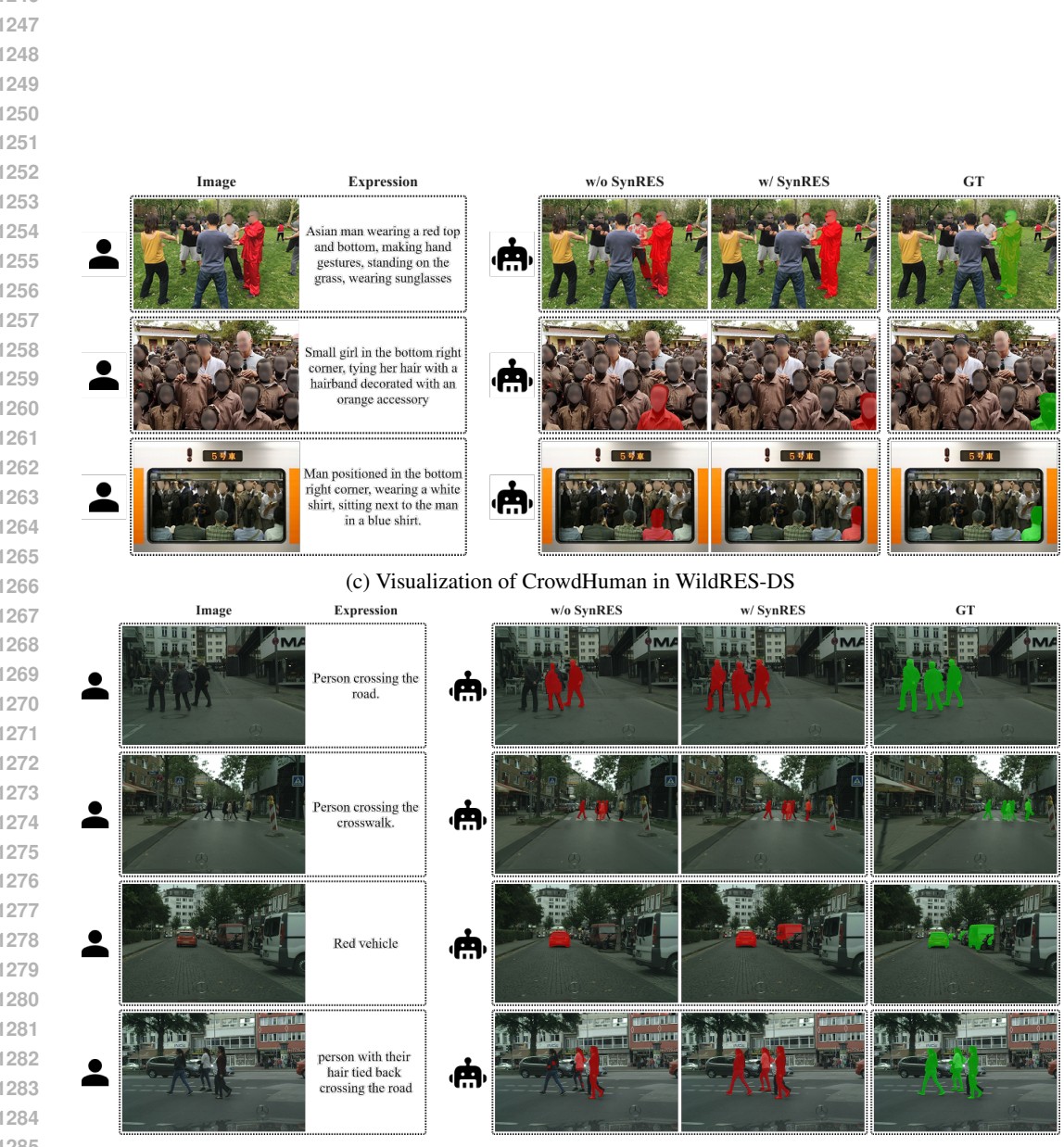

(c) Visualization of CrowdHuman in WildRES-DS

(d) Visualization of Cityscapes in WildRES-DS

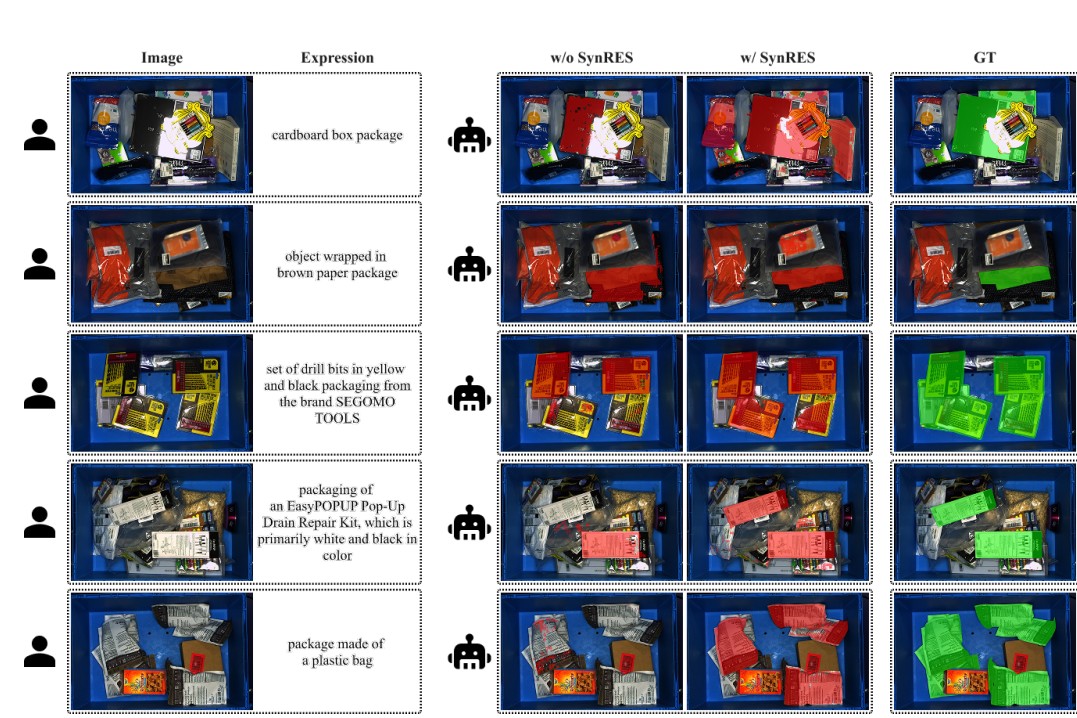

(e) Visualization of ARMBench in WildRES-DS

Figure E: More sampled example from WildRES dataset.

