# OpenReview forum: "Towards Robust Referring Expression Segmentation for Complex Reasoning in the Wild"
_ICLR.cc/2026/Conference — Submitted to ICLR 2026_

### Official Review · Reviewer_2H4Y · 2025-10-26

**Soundness:** 3
**Presentation:** 3
**Contribution:** 2
**Rating:** 4
**Confidence:** 4

**Summary:**

This paper introduces WildRES, a new benchmark designed to evaluate Referring Expression Segmentation (RES) models under complex, real-world conditions that require advanced compositional reasoning. Unlike existing benchmarks, which mainly focus on simple, single-target expressions with limited attributes, WildRES includes long, attribute-rich expressions, multi-target cases with shared attributes, and domain-shifted images from scenes such as crowds, driving, and robotics. To address the lack of large-scale annotated data for such complex scenarios, the authors further propose SynRES, an automated synthetic data generation pipeline that creates densely paired image–mask–text triplets through dense caption-driven synthesis, semantic alignment grouping, and domain-aware augmentations. Experiments across several strong RES baselines (LISA, GSVA, and GLaMM) show that incorporating SynRES significantly improves performance and generalization on both WildRES and traditional datasets like RefCOCO, RefCOCO+, and RefCOCOg.

**Strengths:**

1. A novel benchmark for RES is proposed, which is meaningful and valuable.
2. A new pipeline is proposed to generate and augment data that can be used for training RES models.
3. Experimental results demonstrate the effectiveness of the proposed SynRES.

**Weaknesses:**

1. The WildRES benchmark only has 724 images, which is relatively small.
2. In SynRES, why are the referring expressions synthesized and expanded from real images? Would it be possible to let an LLM directly generate a large number of referring expressions instead? This might further enhance the diversity of the dataset.
3. In Section 4.3, the authors claim that the proposed method can help mitigate gender bias. However, no experiments are provided to support this claim. If the proposed synthetic dataset truly reduces bias, it would represent an interesting and valuable contribution.
4. In Tables 1 and 2, all experiments are conducted using MLLM-based segmentation models. Can the proposed data synthesis approach also help enhance the performance of traditional RES models that do not rely on LLMs?
5. As indicated in Lines 373-374, the experiments are conducted by fine-tuning models that have already been pretrained on segmentation data. It would be interesting to further verify whether the proposed synthetic data could also be used to train these models from scratch.
6. As shown in Table H of the appendix, the performance improves only slightly when the amount of synthetic training data increases from 25% to 100%. Does this indicate that the proposed synthetic-data–based training strategy lacks strong scalability?

**Questions:**

1. During fine-tuning, apart from the proposed SynRES dataset, why do the authors also use other datasets such as gRefCOCO? Is the synthetic data alone insufficient for training? This raises another concern: is the observed performance improvement truly due to the synthetic data, or simply because tarining on existing datasets for more steps?
2. In Table G of the appendix, there is a sharp performance fluctuation when $\tau$ increases from 0.6 to 0.65. Could the authors provide an explanation for this behavior?
3. Does WildRES include reasoning-required cases, similar to those in the ReasonSeg benchmark?

I would greatly appreciate it and would be willing to reconsider my score if the authors address my concerns and questions.

---

> ### Author Response · Authors · 2025-11-20
> **Response to Reviewer 2H4Y - 1**
>
> We sincerely thank Reviewer 2H4Y for the valuable and detailed feedback. Your comments have greatly helped improve the clarity and completeness of our work. We have addressed your concerns regarding benchmark size, data synthesis, bias mitigation, scalability and other concerns below.
>
> > [W1] Benchmark size (724 images) is relatively small
>
> We believe that WildRES effectively benchmarks challenging scenarios for existing RES models. Its scale is also aligned with well-established diagnostic CV benchmarks where quality and difficulty outweigh dataset size, such as Winoground (CVPR 2022, 400 examples, 580+ citations), Visual Riddles (NeurIPS 2024, 400 images), HallusionBench (CVPR 2024, 346 images), and VLR-Bench (COLING 2025, 300 samples). WildRES is designed as a diagnostic evaluation benchmark where quality and coverage of challenging scenarios are paramount, not dataset scale. Our benchmark's value lies in its qualitative distinctiveness: long, attribute-rich queries and non-distinctive multi-target scenarios across diverse domains. The consistent performance degradation across models compared to RefCOCO(+/g) and ReasonSeg (779 test images) validates WildRES's diagnostic power in revealing fundamental capability gaps in complex reasoning scenarios, demonstrating that 724 carefully curated images effectively expose model limitations.
>
> > [W2] Why anchor to real images instead of direct LLM generation?
>
> Real-image anchoring enables domain-aware augmentation in Section 4.3 by pairing real and synthetic samples within mosaic compositions, thereby reducing the domain gap. It also preserves semantic consistency across image–mask–text triplets—something that LLM-only generation cannot ensure. First, our captioning-based generation creates a direct lineage from real to synthetic data. Because expressions are extracted from real images with ground-truth masks (Step 1), we seamlessly blend original real images and their synthetic counterparts within the same mosaic (Step 3), exposing the model to real-synthetic pairs simultaneously while leveraging reliable ground-truth annotations to supervise both domains. Crucially, deriving multiple expressions from the same real object enables densely paired compositional synthetic training where diverse descriptions refer to identical masks, providing rich positive supervision without hard negatives. In contrast, LLM-only generation creates semantic misalignment: for a "man" label, LLMs may independently generate "man with blue shirt" vs. "man with white shirt," causing conflicting attributes when paired with shared masks in mosaics, preventing dense pairing and introducing inconsistencies. Empirically, adding synthetic data without real-image grounding (FreeMask) degrades performance, while Table 3 validates Step 2's importance (gIoU: 41.3→39.1 without it), demonstrating that real-image anchoring balances diversity and triplet reliability for RES.
>
> > [W3] Can authors provide experimental evidence for mitigating gender bias?
>
> Yes—SynRES reduces gender sensitivity by 3× (0.4 vs 1.1 point gap), demonstrating empirical bias mitigation on WildRES. We measured gIoU changes when replacing gender-specific expressions with gender-neutral alternatives on LISA-7B: "man/woman" → "person" and "boy/girl" → "child":
>
> | Method              | Original (man/woman) | Gender-neutral (person) | Gap (↓) |
> |---------------------|----------------------|--------------------------|---------|
> | LISA-7B w/ SynRES   | 33.1                 | 32.7                     | 0.4     |
> | LISA-7B w/o SynRES  | 31.4                 | 30.3                     | 1.1     |
>
> Our method (with SynRES) shows only 0.4 point variation, demonstrating counterfactual consistency. In contrast, the baseline (w/o SynRES) exhibits 1.1 point fluctuation, indicating 3× higher gender sensitivity. This smaller gap proves SynRES-trained models rely less on gender cues and more on visual attributes, successfully mitigating gender bias. We added these results to Section 5.3.3.

---

> ### Author Response · Authors · 2025-11-20
> **Response to Reviewer 2H4Y - 2**
>
> > [W4] Does SynRES also improve traditional non-LLM RES models?
>
> Yes—SynRES consistently improves traditional RES models, achieving +3.6 gIoU average improvement on WildRES-DS (21.6→25.2) with ReLA, a representative cross-modal fusion architecture. We expanded our evaluation to include ReLA, which we selected because it is designed to address long sentences and multiple targets found in the gRefCOCO dataset (unlike standard models tailored for RefCOCO/+/g), making it the most suitable candidate to benchmark against the complex scenarios of WildRES. The detailed results are shown below:
>
> | Model | Training Data | Val (Overall) | Test (Overall) | Test (Many Attr) | Test (Shared Attr) |
> |-------|---------------|---------------|----------------|------------------|---------------------|
> | ReLA | gRefCOCO Only  | 30.0 / 33.6 | 25.9 / 28.0 | 15.6 / 15.6 | 36.9 / 37.9 |
> | ReLA | + SynRES (Ours) | **33.9 / 38.1** | **27.8 / 31.7** | **18.2 / 19.9** | **38.0 / 39.8** |
>
> | Model | Training Data | CrowdHuman | Cityscapes | ARMBench | Average |
> |-------|---------------|------------|------------|----------|---------|
> | ReLA | gRefCOCO Only  | 14.4 / 13.8 | 23.5 / 40.3 | 26.9 / 24.4 | 21.6 / 26.2 |
> | ReLA | + SynRES (Ours) | **21.4 / 20.7** | **25.9 / 40.5** | **28.4 / 27.0** | **25.2 / 29.4** |
>
>
> Our analysis highlights that typical RES models like ReLA achieve only 15.6 gIoU in "Many Attribute" scenarios—significantly lower than MLLM-based models (e.g., LISA-7B) due to the limited capacity of the BERT encoder. However, SynRES effectively mitigates this bottleneck, boosting "Many Attribute" performance and proving its efficacy across different architectures beyond MLLMs. We have included these experimental results and analysis in the revised manuscript.
>
> > [W5] Does synthetic data work for scratch training beyond fine-tuning?
>
> We are conducting this experiment—preliminary results from ReLA (a traditional RES model showing substantial improvements with SynRES, +3.6 gIoU on WildRES-DS) strongly suggest that from-scratch training should also benefit. However, training MLLM-based models (e.g., LISA) from scratch requires significantly longer computation time (ETA >72 hours) compared to fine-tuning experiments. We are prioritizing completion of other experimental validations first and will report from-scratch training results in the rebuttal if the experiment completes in time, or include them in the future version. We believe SynRES's effectiveness for from-scratch training is highly promising given its consistent improvements across diverse architectures (MLLM-based and traditional cross-modal fusion models) and training scenarios.
>
> > [W6] Does performance plateau indicate lack of scalability?
>
> As studied in ealrier work [1], the scalability is highly correlated with the diversity of the synthetic data. Current results indicate that scalability is constrained by the capacity of existing text-to-image models to generate sufficiently diverse concepts, particularly in multi-target scenarios. Synthetic images scale less effectively than real data because off-the-shelf text-to-image models cannot generate certain concepts, significantly impairing downstream model training [2]. Current T2I models including SANA are inherently object-centric, struggling to produce compositionally diverse multi-target scenes that RES requires. Scaling from 25% to 100% synthetic data causes saturation on available single-object variations rather than richer multi-target interactions. However, this limitation is temporary rather than fundamental. Purely synthetic training can rival real-data performance with sufficiently capable generative models, and better generative models in the future will further enhance effectiveness [3]. Our WildRES-DS cross-domain results validate that our framework's core design—densely paired compositional training with domain-aware augmentation—scales effectively when diverse scenarios exist. As T2I models advance to generate truly diverse multi-target compositions, our approach will naturally inherit these improvements.
>
> [1] Chen et al. "On the diversity of synthetic data and its impact on training large language models." arXiv preprint arXiv:2410.15226 (2024).
>
> [2] Fan et al., "Scaling Laws of Synthetic Images for Model Training ... for Now," CVPR 2024.
>
> [3] Tian et al., "Learning Vision from Models Rivals Learning Vision from Data," CVPR 2024.

---

> ### Author Response · Authors · 2025-11-20
> **Response to Reviewer 2H4Y - 3**
>
> > [Q1.1] Why mix real data with synthetic data?
>
> Real data is necessary because synthetic data alone degrades performance (37.1→35.5 gIoU) due to inherent domain gaps in texture, color tone, and background artifacts. Our ablation study (Table 3) empirically validates this: training with only real data achieves 37.1 gIoU, while adding synthetic data without mosaic augmentation drops performance to 35.5 gIoU. This degradation demonstrates that raw synthetic data alone harms performance. However, with mosaic augmentation blending real and synthetic images (1 real + 3–8 synthetic), performance jumps to 41.3 gIoU, proving that real images serve as domain adaptation anchors bridging the synthetic-to-real gap. Examples in Appendix Figure C show that current T2I models (SANA) generate images distinguishable from real ones despite semantic correctness.
>
> > [Q1.2] Improvement from synthetic data or just more training steps?
>
> Improvements stem definitively from synthetic data quality, not training duration—all methods use identical 5,000 training steps, and FreeMask synthetic data degrades performance (34.5→31.6 gIoU, L458-461), proving “more synthetic data” ≠ better results. All experiments in Tables 1–2 use identical training protocols (*i.e.*, with the same training steps) from official pretrained weights with WildRES-ID validation, ensuring fair comparison. If gains were from training duration, FreeMask should improve, not degrade performance. SynRES’s three innovations explain the difference: (1) dense caption-driven synthesis for attribute-rich triplets, (2) Image-Text Aligned Grouping ensuring semantic consistency (ablation: 41.3→40.1 without it), and (3) domain-aware augmentations. Computationally, SynRES generation requires only 73 GPU-hours with 4 GPUs for reusable datasets. Therefore, improvements stem from systematic data quality, not computational budget or training duration.
>
> > [Q2] Table G, why is there a sharp performance fluctuation when $\tau$ increases from 0.6 to 0.65 regarding grouping threshold $\tau$?
>
> The observed fluctuation at $\tau$=0.6 represents stochastic training variance rather than systematic instability. To verify this, we conducted three additional experiments with different random seeds at $\tau$=0.6, yielding consistent results of 40.4, 40.7, and 40.8 gIoU—all substantially higher than the baseline w/o SynRES (37.1 gIoU). The original lower value 39.6 at $\tau$=0.6 appears to be an outlier resulting from stochastic factors inherent to the training process (e.g., batch sampling, mosaic augmentation). This is also comparable to the stable range of $\tau\in$[0.65,0.75] (40.2–41.3 gIoU), demonstrating robust and consistent improvement across the hyperparameter range. Moreover, this variance level is substantially lower than those reported in comparable synthetic segmentation methods, which show sensitivity to hyperparameters such as a prune threshold $\alpha$ that yields a performance fluctuation of about 6.3 points mIoU (e.g., 83.2%–89.5% in [4]), with similar patterns found in [5], [6].
>
> [4] Z. Wu et al., "DiffuMask: Synthesizing Images with Pixel-level Annotations for Semantic Segmentation Using Diffusion Models," ICCV 2023.
>
> [5] J. Zhang et al., "SegGen: Supercharging Segmentation Models with Text2Mask and Mask2Img Synthesis," ECCV 2024.
>
> [6] S. Minaee et al., "FreeMask: Synthetic Images with Dense Annotations Make Stronger Segmentation Models," NeurIPS 2023.
>
> > [Q3] Does WildRES include reasoning-required cases, similar to those in the ReasonSeg?
>
> Reasoning-required--yes, simliar to Reasonseg--No. While WildRES includes cases requiring explicit fine-grained compositional reasoning, ReasonSeg focuses on implicit, world-knowledge–driven queries that require indirect reasoning to distinguish fine-grained differences. Specifically, WildRES requires attribute-rich expressions (11+ attributes) for Many Attribute cases, evaluating compositional understanding through explicit visual grounding (e.g., "the person in the bottom right corner, wearing the beige crossbody bag, taking a photo with his one hand"). Additionally, Shared Attribute cases test the ability to tell apart multiple objects that share similar features, requiring the model to notice small differences when attributes apply to several items in the scene.
>
> ---
> Thank you for your careful consideration of our work and our response. We welcome the opportunity to discuss any remaining questions or issues you might have.

---

> > ### Comment · Reviewer_2H4Y · 2025-11-22
> > **Thank authors for the response**
> >
> > Thank authors for the detailed responses. Most of my cocnerns and questions have been well addressed, so I increse my score. I would be happy to further increase my score if the results of from-scratch training experiments on LISA become available and demonstrate the effectiveness of the proposed method.

---

> ### Author Response · Authors · 2025-11-27
> **Thank you for your positive feedback**
>
> We sincerely thank you for your positive feedback and score increase to 6, as well as for helping us strengthen our manuscript. We are actively conducting the training-from-scratch experiment and will do our best to complete it within the rebuttal window; should it not be finished in time due to the high computational cost, we are fully committed to incorporating these results in our future work.

---

> ### Author Response · Authors · 2025-12-02
>
> We greatly appreciate your constructive engagement and are grateful for your increased score to 6.
>
>
> Regarding the from-scratch training experiment, we have actively attempted to reproduce baseline LISA and apply SynRES as requested. Our training attempts from scratch using the official LISA code consistently produced extremely low performance metrics (gIoU and cIoU < 0.1), failing to approach the reported baseline results. We subsequently discovered this is a well-documented reproducibility challenge across the community, with multiple researchers reporting identical metric mismatches in [LISA Issue #135](https://github.com/dvlab-research/LISA/issues/135). We are actively debugging and re-running experiments with various configurations, but establishing a fair baseline requires resolving these reproducibility issues in addition to substantial computational cost (>300 GPU hours). Given these widespread technical barriers and limited time in the rebuttal period, completing a full, reliable training cycle is challenging.
>
>
> Nevertheless, we remain fully committed to this investigation. We will continue these experiments beyond the rebuttal phase to further validate SynRES's effectiveness in from-scratch settings and will incorporate the final results into the future releases.
>
> Thank you again for your valuable insights which have significantly strengthened our paper.

---

### Official Review · Reviewer_yvJL · 2025-10-28

**Soundness:** 2
**Presentation:** 3
**Contribution:** 3
**Rating:** 6
**Confidence:** 5

**Summary:**

Current Referring Expression Segmentation benchmarks mainly focus on either single targets with short queries or multiple targets from distinctly different queries on a single domain. To address this issue, this paper proposes WildRES, which incorporates long queries with diverse attributes and non-distinctive queries for multiple targets. Therefore, the proposal benchmark aims to deal with complex reasoning in a real-world setting. In light that existing models perform significantly worse on WildRES, this paper proposes SynRES, an automatic pipeline to generate paired synthetic data. SynRES shows promising performance gain for different models, not only on the WildRES dataset, but also on the classic RES benchmarks.

**Strengths:**

The WildRES is well curated, which complements the current limitation in the RES benchmarks in 1) providing more attributes for the  single object referring, 2) images featuring multiple objects sharing the same class but distinct attributes


It is relatively hard to obtain high-quality paired data for the RES tasks, and the motivation to leverage the automatic synthetic data makes sense. The synthetic data generation pipeline is helpful in generating data in a more controlled setting: many attributes/ identical objects with different attributes. The author proposes a solid pipeline to conduct the data generation: 1) captioning to generate diverse attributes and T2I generation for the synthetic images; 2) pseudo-label generation and filtering via group similarity; 3) RES-specific augmentation enhancement. Comprehensive ablation studies are provided to justify the key design choices in this synthetic data generation workflow.

Experimental results across both conventional RES benchmarks and the proposed WildRES dataset—under in-domain and cross-domain settings—demonstrate strong performance and validate the effectiveness of the proposed approach.

**Weaknesses:**

The method part for the SynRES can be organized in a better way. In Section 4.2, it is good to have clear mathematical annotations, but it would be better to organize them in a more systematic manner. It is also suggested to have some high-level ideas before diving into the technical details. For example, what is the high-level idea and assumption for clustering according to expression pairs' similarity in a global manner (i.e., averaging over images)? Without it, the design choice would seem to be more random and not well-motivated.

In the Debiased Text Augment in Section 4.3, I understand that if you do the superclass replacement for the conventional RES dataset, there can be false negatives present. It also makes sense that we can avoid it with the synthetic data, but the way that it is achieved is not obvious enough. The author should make this part clearer.

From the LISA baseline in Table one,  it seems that the improvement on the multiple objects with shared attributes is the most significant; this also applies to the LISA in Table 2 for the WildRES-DS setting. Adding the synthetic data training seems to hurt the many attribute performance a little bit in terms of gIoU in some settings. Can the Author comment on the possible reason?

**Questions:**

See above

---

> ### Author Response · Authors · 2025-11-20
> **Response to Reviewer yvJL - 1**
>
> We sincerely thank Reviewer yvJL for the valuable and detailed feedback. Your comments have greatly helped improve the clarity and completeness of our work. We have addressed your concerns about method presentation, text augmentation clarity, and performance variations
>
> > [W1] Missing high-level ideas before mathematical details in method section
>
> We added high-level intuition paragraphs at the beginning of Sections 4.2 and 4.3 in the revised version to address this concern. You are absolutely correct that Section 4.2 would benefit from presenting the motivation and assumptions before diving into mathematical details.
>
> Section 4.2 now begins with: "T2I models are inherently stochastic, generating $m$ images from identical prompts with different seeds to produce visual variations where not all capture each expression feature. We leverage multi-view consensus by computing mIoU globally across images to aggregate independent observations and filter outliers. Expression pairs exceeding threshold~$\tau$ form consensus groups whose masks are refined through averaging, yielding robust pseudo-masks."
>
> Section 4.3 now begins with: "Step 3 addresses synthetic data limitations: isolated single-object images lack multi-target complexity, and models overfit to specific categories. Mosaic augmentation creates multi-target scenes while reducing synthetic-real domain gaps. Superclass replacement (e.g., woman$\rightarrow$person, probability~$p$) shifts focus from categories to attributes, enabling robust vocabulary generalization."
>
> These additions provide the high-level design rationale and assumptions before presenting the technical implementation details, making the design choices well-motivated rather than seemingly random.
>
> > [W2] Clarify how synthetic data avoids false negatives in debiased text augmentation
>
> Synthetic data avoids false negatives through verbose expressions (averaging 11.4 words) that preserve sufficient discriminative cues after superclass replacement, unlike short RefCOCO expressions (averaging ~3.6 words). Crowdsourced RefCOCO expressions provide minimal cues—for example, "woman" becomes problematic when replaced with "person" if a "man" exists. In contrast, SynRES expressions like "woman with hair tied up in a bun in green striped jumper" → "person with hair tied up in a bun in green striped jumper" preserve 10+ words (hairstyle, clothing) that uniquely identify the target despite superclass replacement. We enhanced Section 4.3 to explicitly clarify this length-based mechanism.
>
> > [W3] Why adding synthetic data hurts MA performance in some settings?
>
> The MA performance degradation observed in GSVA-7B and LISA-7B (−0.2 to −0.7 gIoU) largely stems from the limited expressive capacity of the Rank-8 LoRA in their LLM components when processing MA’s lengthy, attribute-rich queries (11+ words). In contrast, SA tasks—being shorter and less linguistically complex—benefit substantially within the same capacity limits, achieving improvements of +4.3 to +5.9 gIoU. Larger models (e.g., LISA-13B) and GLaMM—pretrained on the substantially larger 7.5M-concept GranD dataset—offer greater model capacity, which alleviates this limitation and enables performance gains on MA (+0.9 to +3.2 gIoU). Overall, performance consistently increases in LISA-7B (+2.0 in WildRES-ID to +3.8 gIoU in WildRES-DS), demonstrating SynRES effectiveness despite this minor MA trade-off in capacity-constrained settings.
>
> ---
> Thank you for your careful consideration of our work and our response. We welcome the opportunity to discuss any remaining questions or issues you might have.

---

> ### Author Response · Authors · 2025-11-27
> **A Kind Reminder**
>
> Thank you for your time and effort in reviewing our paper. We would like to kindly remind you of our response and welcome any additional questions or clarifications you may have regarding our work.

---

### Official Review · Reviewer_qvxs · 2025-10-30

**Soundness:** 2
**Presentation:** 2
**Contribution:** 2
**Rating:** 4
**Confidence:** 4

**Summary:**

This paper focuses on the Referring Expression Segmentation (RES) task. To address the issues of limited evaluation capability in existing benchmarks and suboptimal performance of models in real-world scenarios, this paper proposes the WildRES benchmark dataset and the SynRES synthetic data generation and augmentation pipeline, and validates the effectiveness of the proposed approach.

**Strengths:**

1. WildRES addresses the evaluation of complex real-world scenarios in the RES field and can serve as a new standard for measuring models' complex reasoning abilities.

2. Specifically, SynRES addresses the shortage of complex RES data at a low cost by automatically generating high-quality, densely annotated paired data. This pipeline can enhance model performance in complex queries and cross-domain scenarios, while remaining compatible with classic benchmarks.

**Weaknesses:**

1. The mentioned issues include the lack of evaluation in complex scenarios, poor generalization to complex scenarios, and overly brief descriptions. However, these issues have been discussed in many existing RES or RIS benchmarks, such as MMR[1], LLM-Seg40K[2], ReasonSeg[3], MUSE[4], etc. Moreover, these benchmarks are more complex than the one proposed by the authors. The authors need to clearly explain the differences and advantages between the WildRES benchmark dataset and these existing benchmarks.

2. The experiment only selects three open-source models (LISA, GSVA, and GLaMM) and fails to cover other typical RES architectures (e.g., lightweight Transformer-based models or cross-modal fusion models), which fails to fully demonstrate the generalizability of SynRES across different architectures. Additionally, the authors do not compare its performance with that of other existing models (such as [5][6][7]) on the RefCOCO series datasets. Furthermore, WildRES-DS only covers three types of scenarios, and this benchmark fails to truly achieve generalization to out-of-domain data.

3. The SynRES synthetic data pipeline proposed by the authors is a combination of two types of data augmentation. Essentially, it merges Mosaic augmentation with text rewriting augmentation, both of which are already widely used in the RES field. The authors need to clarify the novelty of this pipeline.

4. The authors obtain better test results by using more training data, rendering this comparison unfair. It is necessary to compare the speed and efficiency of this method with other methods—specifically in terms of training time and FLOPs—to determine if there are any differences.

[1] MMR: A Large-scale Benchmark Dataset for Multi-target and Multi-granularity Reasoning Segmentation[C]//The Thirteenth International Conference on Learning Representations.

[2] Llm-seg: Bridging image segmentation and large language model reasoning[C]//Proceedings of the IEEE/CVF Conference on Computer Vision and Pattern Recognition. 2024: 1765-1774.

[3] Lisa: Reasoning segmentation via large language model[C]//Proceedings of the IEEE/CVF Conference on Computer Vision and Pattern Recognition. 2024: 9579-9589.

[4] Pixellm: Pixel reasoning with large multimodal model[C]//Proceedings of the IEEE/CVF Conference on Computer Vision and Pattern Recognition. 2024: 26374-26383.

[5]  Visionllm v2: An end-to-end generalist multimodal large language model for hundreds of vision-language tasks[J]. Advances in Neural Information Processing Systems, 2024, 37: 69925-69975.

[6] Vltp: Vision-language guided token pruning for task-oriented segmentation[C]//2025 IEEE/CVF Winter Conference on Applications of Computer Vision (WACV). IEEE, 2025: 9353-9363.

[7]  F-lmm: Grounding frozen large multimodal models[C]//Proceedings of the Computer Vision and Pattern Recognition Conference. 2025: 24710-24721.

**Questions:**

1. WildRES only covers three types of cross-domain scenarios: dense crowds (CrowdHuman), autonomous driving (Cityscapes), and robotics (ARMBench) (Section 3.2). What is the main rationale for the authors choosing these three types of scenarios? Does it verify whether the domain gaps in these three scenarios can represent the real cross-domain requirements of RES?

2. The paper only focuses on segmentation accuracy (gIoU/cIoU) and does not test the training efficiency of the model enhanced by SynRES—whether adding synthetic data for training affects the training speed—and should conduct a more thorough performance comparison with existing RES models.

3. SynRES's text augmentation uses hypernym replacement (e.g., woman→person, Section 4.3), with a default replacement probability of p = 0.7 (Section 5.1). However, the authors need to explain whether hypernym replacement might cause semantic ambiguity when the class noun in the expression is strongly associated with an attribute (e.g., pregnant woman → person, where the attribute "pregnant" is tightly bound to "female"). It is necessary to clarify the adaptive differences under different replacement probabilities, and if replacement leads to semantic ambiguity, whether it could introduce new segmentation errors.

---

> ### Author Response · Authors · 2025-11-20
> **Response to Reviewer qvxs - 1**
>
> We sincerely thank Reviewer qvws for the valuable and detailed feedback. Your comments have greatly helped improve the clarity and completeness of our work. We have addressed your concerns regarding benchmark complexity, model coverage, methodological novelty and other concerns below.
>
>
> > [W1] Existing benchmarks are more complex than proposed WildRES
>
> WildRES differentiates itself from existing benchmarks (e.g., MMR, ReasonSeg) by addressing two critical complexities they overlook: attribute confusion in single targets and shared-attribute ambiguity for multiple targets.
>
> **First, attribute confusion with single targets:** When single targets are described with many attributes (e.g., Fig 6 row 2 "the person in the bottom right corner, wearing the beige crossbody bag, taking a photo with his one hand"), our experiments consistently show that current models incorrectly match objects sharing only partial attributes (e.g., Fig 6 row 2 col 2 attribute "bottom right corner"). Existing benchmarks do not systematically evaluate this failure mode.
>
> **Second, shared-attribute ambiguity for multiple targets:** WildRES uniquely evaluates non-distinctive multi-target scenarios where multiple objects share similar attributes without explicit disambiguation mechanisms (e.g., red vehicles on the road")—requiring models to identify ALL matching instances rather than selecting among pre-distinguished alternatives. This capability is essential for autonomous driving and robotic manipulation but remains unaddressed.
> Our experiments reveal significant performance degradation on WildRES (particularly WildRES-DS with domain shift), demonstrating that current methods optimized for distinctive-target benchmarks fail to generalize to these practical, ambiguous scenarios.
>
> > [W2.1] Limited coverage of typical RES architectures
>
> In response to the request, we have expanded our evaluation to include ReLA, a representative cross-modal fusion RES architecture. We specifically selected ReLA because it is designed to address the long sentences and multiple targets found in the gRefCOCO dataset (unlike standard models tailored for RefCOCO/+/g), making it the most suitable candidate to benchmark against the complex scenarios of WildRES. Our experiments confirm that SynRES consistently improves performance, particularly by compensating for the limitations of lightweight text encoders in complex scenarios. The detailed results are shown below:
>
> | Model | Training Data | Val (Overall) | Test (Overall) | Test (Many Attr) | Test (Shared Attr) |
> |-------|---------------|---------------|----------------|------------------|---------------------|
> | ReLA | gRefCOCO Only  | 30.0 / 33.6 | 25.9 / 28.0 | 15.6 / 15.6 | 36.9 / 37.9 |
> | ReLA | + SynRES (Ours) | **33.9 / 38.1** | **27.8 / 31.7** | **18.2 / 19.9** | **38.0 / 39.8** |
>
> | Model | Training Data | CrowdHuman | Cityscapes | ARMBench | Average |
> |-------|---------------|------------|------------|----------|---------|
> | ReLA | gRefCOCO Only  | 14.4 / 13.8 | 23.5 / 40.3 | 26.9 / 24.4 | 21.6 / 26.2 |
> | ReLA | + SynRES (Ours) | **21.4 / 20.7** | **25.9 / 40.5** | **28.4 / 27.0** | **25.2 / 29.4** |
>
>
>
> Our analysis highlights the architectural limitation of typical RES models like ReLA, which achieves only 15.6 gIoU in "Many Attribute" scenarios—significantly lower than MLLM-based models (e.g., 30.1 gIoU in LISA-7B) due to the limited capacity of the BERT-based language encoder. However, SynRES effectively mitigates this bottleneck, boosting "Many Attribute" performance and improving average domain generalization on WildRES-DS by 3.6 gIoU (from 21.6 to 25.2), proving its efficacy across different architectures. We have included these experimental results and analysis in Section 5 of the revised manuscript.
>
> > [W2.2] Lack of comparison with existing models on RefCOCO series datasets
>
> We added VisionLLMv2 and F-LMM comparisons to Table E, expanding our baseline coverage on RefCOCO series datasets. VLTP was not included because it does not report RES tasks on RefCOCO series, making direct comparison infeasible. In summary, our method achieves substantial improvements on WildRES, which demands more complex reasoning, while delivering comparable or slightly higher performance on the RefCOCO series.

---

> ### Author Response · Authors · 2025-11-20
> **Response to Reviewer qvxs - 2**
>
> > [W2.3, Q1] Limited scenario diversity in WildRES-DS (three types) insufficient for out-of-domain generalization
>
> We selected these three domains—CrowdHuman, Cityscapes, and ARMBench—because they represent critical, high-stakes application areas with distinct technical challenges that are complementary and radically different from MSCOCO (the foundation of all existing RES benchmarks). Specifically, CrowdHuman addresses densely crowded and heavily occluded scenarios (averaging 22.6 persons/image) critical for security/surveillance; Cityscapes tackles autonomous driving where perception errors have life-threatening consequences and require comprehensive scene understanding; and ARMBench focuses on robotic manipulation for industrial automation requiring advanced perception. While it is challenging for any benchmark to cover the full spectrum of real-world scenarios, our design goal for WildRES-DS is to provide a representative set of diverse, high-impact cases rather than an exhaustive enumeration of all possible conditions. MSCOCO contains natural images with relatively clean conditions and low object density, whereas WildRES-DS encompasses specialized distributions with extreme conditions absent from traditional benchmarks.
> The empirical evidence validates this substantial domain shift:
>
> | Method           | gRefCOCO-val | WildRES-ID (gIoU) | WildRES-DS (gIoU) | Performance Drop     |
> |------------------|--------------|-------------------|-------------------|---------------------|
> | GSVA w/o SynRES  | 72.6         | 34.8              | 26.9              | -7.9                |
> | GSVA w/ SynRES| **73.6**     | **38.0**          | **32.1**          | **-5.9 (reduced)**  |
>
> This demonstrates: (1) significant performance deterioration confirms substantial domain gap; (2) SynRES mitigates this drop (7.9→5.9), proving effective cross-domain generalization. This performance pattern across SOTA architectures validates our domain selection as both pragmatically representative and technically rigorous for evaluating real-world deployment readiness.
>
> > [W3] Clarify novelty beyond standard mosaic and text augmentation
>
> SynRES's novelty lies in automated end-to-end synthetic triplet generation with semantic alignment-based quality assurance, fundamentally differing from conventional augmentation that transforms existing annotated data. Unlike prior augmentation techniques that rely on high-quality annotations or construct mosaics by randomly combining mini-batch samples, SynRES generates mosaics using domain-aware and densely paired compositional synthetic data, along with reliable pseudo-masks obtained via Image–Text Aligned Grouping without requiring annotations. This enables large-scale training data generation while preserving high annotation fidelity. SynRES is an end-to-end automated pipeline that synthetically generates complete image-mask-expression triplets from scratch, addressing the critical challenge of annotation scarcity in complex reasoning scenarios. Our three key innovations work synergistically: (1) Dense caption-driven synthesis automatically generates attribute-rich triplets by leveraging foundation models, eliminating manual annotation; (2) Image-Text Aligned Grouping introduces a novel semantic alignment mechanism that systematically detects and corrects caption-pseudo mask inconsistencies—a quality assurance step absent in standard augmentation that ensures generated masks accurately correspond to textual descriptions; and (3) Domain-aware augmentations, including superclass replacement (which shifts model focus from specific categories to distinguishing attributes, enhancing generalization) and mosaic composition (which creates compositional complexity beyond simple spatial transformation). Critically, our ablation studies demonstrate that removing the semantic alignment component causes significant performance degradation, validating that mask quality assurance is essential.

---

> ### Author Response · Authors · 2025-11-20
> **Response to Reviewer qvxs - 3**
>
> > [W4.1] Better results from more data renders comparison unfair
>
> Adding 400k+ FreeMask synthetic samples to LISA-7B actually reduces performance, directly refuting the concern that our improvements arise merely from using more data. The gains instead come from how the data are generated (Sec. 3.1) and how they are effectively utilized (Secs. 3.2, 3.3), rather than from data quantity alone (Tables 1 and 2, L458–461). In Tables 1 and 2, all methods use identical training steps with controlled validation via WildRES-ID, ensuring fair comparison. This counterexample proves that "more synthetic data" does not automatically improve results. SynRES's improvements stem from data quality through three innovations: (1) dense caption-driven synthesis generating attribute-rich triplets, (2) Image-Text Aligned Grouping ensuring semantic consistency, and (3) domain-aware augmentations enhancing generalization.
>
> > [W4.2, Q2] Compare computational efficiency (training time, FLOPs) with baselines
>
> SynRES offers computational efficiency competitive to both synthetic data generation and fine-tuning. It requires 73 hours on 4 GPUs and performs 5,000 training iterations(only 1/10 the cost of training a model from scratch). Our synthetic data generation takes only 73 hours with 4 GPUs using off-the-shelf models with no additional training costs (Appendix G.3). The generated dataset is reusable across multiple models and experiments, making it highly cost-effective. Also, Because SynRES is a data-centric approach, models trained with and without SynRES have exactly the same training time and FLOPs during fine-tuning.
>
> > Q1
>
> We kindly refer the reviewer to our response under [W2.3, Q1].
>
>
> > Q2
>
> We kindly refer the reviewer to our response under [W4.2, Q2].
>
>
> > [Q3] Explain whether hypernym replacement causes semantic ambiguity and introduces new segmentation errors
>
> Hypernym replacement does not introduce harmful semantic ambiguity or segmentation errors—empirical validation across diverse replacement probabilities (p=0.5~0.9) demonstrates robust performance, and ablation studies confirm it enhances rather than degrades generalization. We clarify that existing models may be biased toward specific query formulations and exhibit limited generalization across hierarchical class relationships expressed in the queries. To address this limitation, our superclass replacement strategy is intentionally designed to shift the model’s focus away from overly specific categorical queries and toward broader, more generalizable concepts—an ability that is essential for robust cross-domain generalization. We acknowledge the reviewer’s concern that a query like “person” may appear biased toward “female” when considering only a single training instance. However, in practice, this bias is naturally mitigated through diverse training examples in which “person” is also associated with “male,” enabling the model to learn a more inclusive and semantically broader representation of the superclass “person”. As a results, our synthetic data generation ensures single-object isolation (as shown in Figure 5), minimizing false negative risks that would arise with such replacements in multi-object real datasets like RefCOCO. Empirical validation confirms effectiveness (Appendix G.1, Table G left):
>
> | Replacement Probability ($p$) | gIoU |
> |----------------------------|------|
> | 0.5 | 40.8 |
> | 0.6 | 40.3 |
> | 0.7 | 41.3 |
> | 0.8 | 41.0 |
> | 0.9 | 40.4 |
>
> p=0.7 achieves optimal performance with robust consistency (gIoU ≥40 across all values). Furthermore, Table 3 ablation demonstrates that removing text augmentation reduces gIoU from 41.3 to 40.1 (-1.2), while eliminating both augmentations drops performance to 35.5—worse than the 37.1 real-only baseline. This proves synthetic data without proper augmentation actually degrades performance due to domain gaps, validating that superclass replacement is essential for effective synthetic data utilization and enhances generalization without introducing segmentation errors.
>
> ---
> Thank you for your careful consideration of our work and our response. We welcome the opportunity to discuss any remaining questions or issues you might have.

---

> ### Author Response · Authors · 2025-11-27
> **A Kind Reminder**
>
> Thank you for your time and effort in reviewing our paper. We would like to kindly remind you of our response and welcome any additional questions or clarifications you may have regarding our work.

---

### Author Response · Authors · 2025-11-20
**General Response**

We sincerely thank the reviewers for their constructive feedback, which has substantially improved this manuscript.

The key changes made based on these comments are highlighted in red within the revised manuscript:

- **High-level intuition added (Reviewer yvJL):** We introduced introductory paragraphs in Sections 4.2 and 4.3 to clarify the motivation and assumptions behind our clustering and augmentation strategies.
- **Additional ReLA experiments (Reviewer qvxs, 2H4Y):** We included experiments with the ReLA model in Section 5 to demonstrate that SynRES benefits general RES architectures, not just MLLM-based models.
- **Gender bias mitigation results (Reviewer 2H4Y):** We added empirical results in Section 5.3.3, showing that SynRES effectively reduces gender sensitivity in segmentation outcomes.
- **Broader baseline comparison (Reviewer qvxs):** The appendix has been updated with new performance metrics for VisionLLM v2 and F-LLM to provide a more comprehensive comparison.
- **Clarifications and refinements:** Throughout the manuscript, we made several adjustments to enhance readability and technical rigor.

Below, we provide detailed, point-by-point responses to each reviewer’s comments. We appreciate their valuable feedback and believe the revisions adequately address their concerns. We are happy to respond to any further questions or suggestions the reviewers may have.

---

### Author Response · Authors · 2025-12-02
**Discussion Summary by Authors**

Dear Area Chair,

We would like to provide a brief summary of the discussions held during the rebuttal period.

Initial reviews raised concerns regarding **the generalizability of SynRES to diverse architectures** and **the validity of our text augmentation pipeline (including potential semantic ambiguity and bias mitigation)**. Additionally, there was a shared question about **whether our performance gains stemmed merely from increased data quantity rather than methodological quality**.

We addressed these points thoroughly in our response. Reviewer **2H4Y** expressed that these concerns were mostly resolved and subsequently **increased their score to 6**. While we did not receive follow-up responses from reviewers **qvxs** and **yvJL**, their concerns regarding **architecture generalizability** and **fair data comparison** closely aligned with those of **2H4Y** and were addressed within the same discussion.

We also carefully responded to all other comments, such as clarifying benchmark comparisons and providing high-level intuition for our method, and have incorporated these improvements into the revised manuscript.

---

### Meta-Review · Area_Chair_GJPn · 2026-01-04

**Summary:**

WildRES is a focused benchmark for long, attribute-rich and non-distinctive multi-target RES, and SynRES is a data-centric synthetic triplet pipeline that consistently improves multiple strong baselines (and a traditional RES model). After rebuttal, the main acceptability question shifts from “does it work?” to “is it sufficiently differentiated/positioned vs prior complex RES benchmarks and is the evaluation scope broad enough.” Considering the concerns about limited novelty/positioning compared to existing benchmarks, the AC learns reject.

**Reviewer Concerns:**

Addressed: added ReLA (non-MLLM) results to support architecture generality; added broader baselines (e.g., VisionLLMv2, F-LLM); clarified WildRES vs prior benchmarks and the rationale for WildRES-DS domains; provided evidence that gains are not just “more data” (FreeMask counterexample + fixed steps) and added efficiency notes; added gender-bias mitigation results and clarified superclass replacement ambiguity.

Still outstanding: novelty/positioning vs existing “complex reasoning” RES/RIS benchmarks may still feel borderline; WildRES-DS domain breadth remains limited by design; from-scratch training validation was not completed (and authors note reproducibility issues).

**Reviewer Scores:**

qvxs (4): likely 4 → 4, overlap/novelty skepticism may persist despite added experiments.
yvJL (6): likely 6 → 6, main requests were presentation/intuition; addressed.
2H4Y (4): 4 → 6, explicitly increased after rebuttal.

---

### Decision · Program_Chairs · 2026-01-26

Reject